# SENSEI: SEMANTIC EXPLORATION GUIDED BY FOUNDATION MODELS TO LEARN VERSATILE WORLD MODELS

## ABSTRACT

Exploring useful behavior is a keystone of reinforcement learning (RL). Intrinsic motivation attempts to decouple exploration from external, task-based rewards. However, existing approaches to intrinsic motivation that follow general principles such as information gain, mostly uncover low-level interactions. In contrast, children's play suggests that they engage in meaningful high-level behavior by imitating or interacting with their caregivers. Recent work has focused on using foundation models to inject these semantic biases into exploration. However, these methods often rely on unrealistic assumptions, such as environments already embedded in language or access to high-level actions. To bridge this gap, we propose SEmaNtically Sensible ExploratIon (SENSEI), a framework to equip model-based RL agents with intrinsic motivation for semantically meaningful behavior. To do so, we distill an intrinsic reward signal of interestingness from Vision Language Model (VLM) annotations. The agent learns to predict and maximize these intrinsic rewards using a world model learned directly from intrinsic rewards, image observations, and low-level actions. We show that in both robotic and video game-like simulations SENSEI manages to discover a variety of meaningful behaviors. We believe SENSEI provides a general tool for integrating feedback from foundation models into autonomous agents, a crucial research direction, as openly available VLMs become more powerful.[1]

## 1 INTRODUCTION

Achieving intrinsically-motivated learning in artificial agents has been a long-standing dream, making it possible to decouple agents' learning from an experimenter manually crafting and setting up tasks.

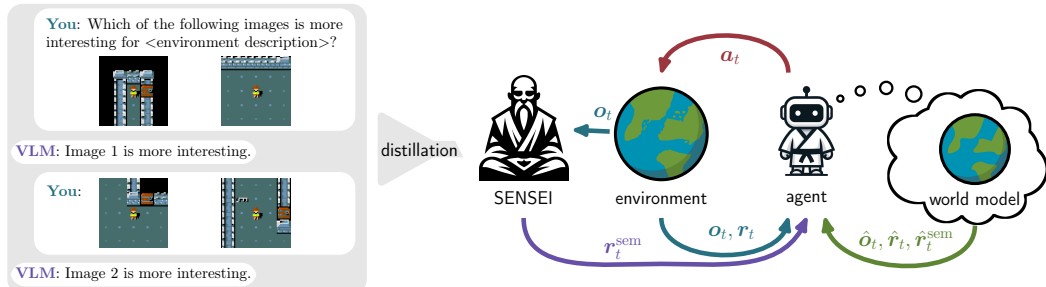

Figure 1: **SENSEI overview**: **(a)** During pre-training we prompt a VLM to compare observations (e.g. images) from an environment with respect to their interestingness. We distill this ranking into a reward function (SENSEI), to guide the exploration of an embodied agent. **(b)** An exploring agent not only receives observations ($o_t$) and rewards ($r_t$) from interactions with the environment but also a semantic exploration reward ($r_t^{\text{sem}}$) from SENSEI. **(c)** The agent learns a world model from its experience to judge the interestingness ($\hat{r}_t^{\text{sem}}$) of states without querying SENSEI.

---

[1]Project website with videos: https://sites.google.com/view/sensei-iclr

Thus, the goal in intrinsically-motivated reinforcement learning (RL) is for agents to explore their environment efficiently and autonomously, constituting a free play phase akin to children's curious play. Various intrinsic reward definitions have been proposed in the literature, such as aiming for state space coverage (Bellemare et al., 2016; Tang et al., 2017; Burda et al., 2019), novelty or retrospective surprise (Pathak et al., 2017; Schmidhuber, 1991), and information gain of a world model (Pathak et al., 2019; Sekar et al., 2020; Sancaktar et al., 2022). However, when an agent starts learning from scratch, there is one fundamental problem: just because something is novel does not necessarily mean that it contains useful or generalizable information for any sensible task (Dubey & Griffiths, 2017).

Imagine a robot facing a desk with several objects. The robot could explore by trying to move through the entire manipulable space or hitting the desk at various speeds. In contrast, human common sense would likely focus on interacting with the objects or drawer of the desk since potential task distributions likely revolve around those entities.

Agents exploring their environment from scratch with intrinsic motivations suffer from a chicken-or-egg problem: *how do you know something is interesting before you have tried it and experienced interesting consequences?* This is a bottleneck for the types of behavior that an agent can unlock during free play. We argue that incorporating human priors into exploration could alleviate this roadblock. Similar points have been raised for children's play. During the first years of life, children are surrounded by their caregivers who ideally encourage and reinforce them while they explore their environment. Philosopher and psychologist Karl Groos has stipulated that there is "a strong drive in children to observe the activities of their elders and incorporate those activities into their play" (Gray, 2017; Groos & Baldwin, 1901).

A potential solution in the age of Large Language Models (LLMs), is to utilize language as a cultural-transmitter to inject "human notions of interestingness" (Zhang et al., 2023a) into RL agents' exploration. LLMs are trained on an immense amount of data produced mostly by humans. Thus, their responses are likely to mirror priors or preferences about what humans find interesting. However, the most prominent works in this domain assume (1) a semantically-grounded environment (Zhang et al., 2023b; Du et al., 2023), (2) in the case of MOTIF (Klissarov et al., 2023), the availability of an offline dataset with exhaustive state-space coverage and messages labeling the unfolding events or (3) access to high-level actions in the embodied environments considered in Zhang et al. (2023a); Du et al. (2023). These assumptions are still detached from the current reality of embodied agents, e.g. in robotics, which don't come with perfect state or event captioners, pre-existing offline datasets nor with robust, abstracted away actions. Furthermore, none of these approaches learn an internal model of "interestingness." Thus, they rely on the LLM, or a distilled module, to continuously guide their exploration and fail to transfer this knowledge to novel states when LLM feedback is not available.

In this work, we propose SEmaNtically Sensible ExploratIon (SENSEI), a framework for Vision Language Models (VLM) guided exploration for model-based RL agents, illustrated in Fig. 1. SENSEI starts with a short description of the environment and a dataset of observations (e.g. images) collected through self-supervised exploration. A VLM is prompted to compare the observations pairwise with respect to their interestingness and the resulting ranking is distilled into a reward function. When the agent explores its environment, it receives semantically-grounded exploration rewards from SENSEI. The agent learns to predict this exploration signal through its learned world model, corresponding to an internal model of "interestingness", and improves its exploration strategy based on these model-based predictions.

Our main contributions are as follows:

- We propose SENSEI, a framework for foundation model-guided exploration with world models.
- We show that SENSEI can explore rich, semantically meaningful behaviors with few prerequisites.
- We demonstrate that the versatile world models learned through SENSEI enable fast learning of downstream tasks.

## 2 METHOD

We consider the setup of an agent interacting with a Partially Observable Markov Decision Process. At each time $t$, the agent performs an action $\boldsymbol{a}_t \in \mathcal{A}$ and receives an observation $\boldsymbol{o}_t \in \mathcal{O}$, composed of an image and potentially additional information. We assume that there exist one or more tasks

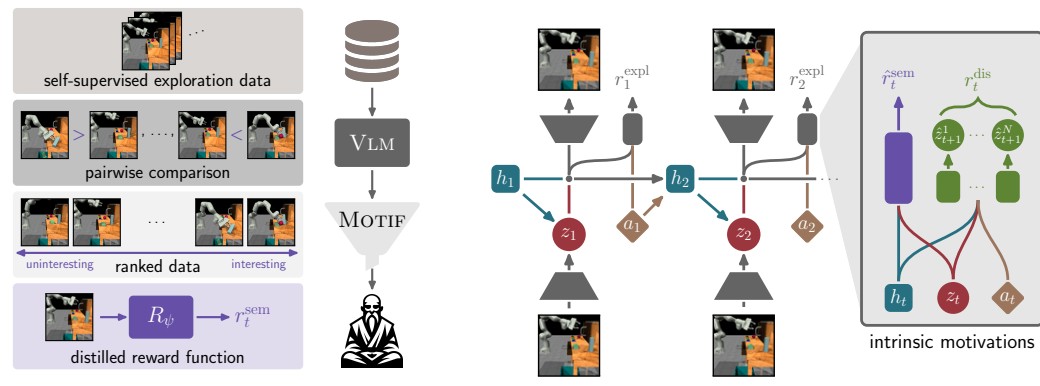

(a) reward function distillation    (b) world model

Figure 2: **Intrinsic rewards in SENSEI**: **(a)** Prior to task-free exploration, we prompt GPT-4 to compare images with respect to the interestingness for a certain environment. From the resulting ranking we distill a reward function $R_\psi$ using VLM-MOTIF. **(b)** Later, an agent learns an RSSM world model from task-free exploration. From each model state, the agent predicts different sources of intrinsic rewards, i.e., epistemic uncertainty-based reward and our distilled semantic reward.

in the environment for which the agent may receive rewards $r_t^{\text{task}} \in \mathbb{R}$ after executing an action. However, during task-free exploration, the agent should select its behavior agnostic to task rewards.

We assume that SENSEI starts with a dataset $\mathcal{D}^{\text{init}} \subset \mathcal{O}$ collected from self-supervised exploration (Sekar et al., 2020; Sancaktar et al., 2022), has access to a pretrained VLM and is provided with short description of the environment, either from a human expert or generated by a VLM, based on some observations from $\mathcal{D}^{\text{init}}$. *Prior* to task-free exploration, SENSEI distills a semantic exploration reward function from VLM annotations (Sec. 2.1). *During* exploration, SENSEI learns a world model (Sec. 2.2) and optimizes an exploration policy through model-based RL and intrinsic reward predictions (Sec. 2.3).

## 2.1 REWARD FUNCTION DISTILLATION: MOTIFate YOUR SENSEI

Prior to task-free exploration, SENSEI needs to distill a semantically grounded intrinsic reward function $R_\psi$ with learnable parameters $\psi$ based on the preferences of a pretrained VLM. While the overall framework of SENSEI is agnostic to the exact distillation method, we chose to use a vision-based extension of MOTIF (Klissarov et al., 2023; illustrated in Fig. 2a), which we refer to as VLM-MOTIF.[2]

MOTIF consists of two phases. In the first phase of **dataset annotation**, the pretrained foundation model is used to compare pairs of observations, creating a dataset of preferences. For this, we prompt the VLM with an environment description and provide pairs of observations from $\mathcal{D}^{\text{init}}$, asking the VLM which image it considers to be more interesting. The annotation function is given by the VLM : $\mathcal{O} \times \mathcal{O} \to \mathcal{Y}$, where $\mathcal{O}$ is the space of observations, and $\mathcal{Y} = \{1, 2, \emptyset\}$ is a space of choices for the first, second or none of the observations. In the **reward training** phase, a reward function is derived from the VLM preferences using standard techniques from preference-based RL (Wirth et al., 2017). A cross-entropy loss function is minimized on the dataset of preference pairs to learn a semantically grounded reward model $R_\psi : \mathcal{O} \to \mathbb{R}$. We use the final semantic reward function $R_\psi$ whenever the agent interacts with its environment: the agent not only receives an observation $o_t$ and reward $r_t$ after executing an action $a_t$, but also receives a semantically-grounded exploration reward $r_t^{\text{sem}} \leftarrow R_\psi(o_t)$ (see Fig. 1, center).

---

[2]Original MOTIF (Klissarov et al., 2023) assumes an environment where there exist captions that describe the events at each time $t$. Thus, they can use LLMs to annotate the captions instead of using VLMs to annotate observations.

## 2.2 World Model: Let your SENSEI dream

We assume a model-based setting, i.e., the agent learns a world model from its interactions. Following DreamerV3 (Hafner et al., 2023), we implement the world model as a Recurrent State Space Model (RSSM) (Hafner et al., 2019b). The RSSM with learnable parameters $\phi$ is computed by

$$\text{Posterior:} \quad \boldsymbol{z}_t \sim q_\phi(\boldsymbol{z}_t \mid \boldsymbol{h}_t, \boldsymbol{o}_t) \tag{1}$$

$$\text{Dynamics:} \quad \boldsymbol{h}_{t+1} = f_\phi(\boldsymbol{a}_t, \boldsymbol{h}_t, \boldsymbol{z}_t) \tag{2}$$

$$\text{Prior:} \quad \hat{\boldsymbol{z}}_{t+1} \sim p_\phi(\hat{\boldsymbol{z}}_{t+1} \mid \boldsymbol{h}_{t+1}) \tag{3}$$

In short, the RSSM encodes all interactions through two latent states, a stochastic state $\boldsymbol{z}_t$ and a deterministic memory $\boldsymbol{h}_t$. At each time $t$, the RSSM samples a new stochastic state $\boldsymbol{z}_t$ from a posterior distribution $q_\phi$ computed from the current deterministic state $\boldsymbol{h}_t$ and new observation $\boldsymbol{o}_t$ (Eq. 1). The RSSM updates its deterministic memory $\boldsymbol{h}_{t+1}$ based on the action $\boldsymbol{a}_t$ and previous latent states (Eq. 2). Next, the model predicts the next stochastic state $\hat{\boldsymbol{z}}_{t+1}$ (Eq. 3). Once the new observation $\boldsymbol{o}_{t+1}$ is received, the next posterior $q_\phi$ is computed and the process is repeated.

Besides encoding dynamics within its latent state, the RSSM is also trained to reconstruct external quantities $y_t$ from its latent state via output heads $o_\phi$:

$$\text{Output heads:} \quad \hat{y}_t \sim o_\phi(\hat{y}_t \mid \boldsymbol{h}_t, \boldsymbol{z}_t) \quad \text{with} \quad y_t \in \{\boldsymbol{o}_t, c_t, r_t, r_t^{\text{sem}}\} \tag{4}$$

The RSSM of DreamerV3 (Hafner et al., 2023) reconstructs observations $\boldsymbol{o}_t$, episode continuations $c_t$, and rewards $r_t$. For SENSEI, we additionally predict the semantic exploration reward $r_t^{\text{sem}}$. The world model is trained end-to-end to jointly optimize the evidence lower bound.

Thus, our world model learns to predict semantic interestingness $\hat{r}_t^{\text{sem}}$ of states (see Fig. 1, right). We could base exploration exclusively on this signal. However, we expect to face many local optima when optimizing for this signal and we do not want to only explore a fixed set of behaviors, but ensure that the agent goes for interesting and yet novel states. To overcome this limitation, Klissarov et al. (2023) post-process $r_t^{\text{sem}}$ and normalize it by episodic event message counts. As we do not assume ground-truth countable event captions, we instead combine our new reward signal with epistemic uncertainty, a quantity that was shown to be an effective objective for model-based exploration (Sekar et al., 2020; Pathak et al., 2017; Sancaktar et al., 2022). Following Plan2Explore (Sekar et al., 2020), we train an ensemble of $N$ models with weights $\{\theta^1, \ldots, \theta^N\}$ to predict the next stochastic latent states with

$$\text{Ensemble predictor:} \quad \hat{\boldsymbol{z}}_t^n \sim g_{\theta^n}(\hat{\boldsymbol{z}}_t^n \mid \boldsymbol{h}_t, \boldsymbol{z}_t, \boldsymbol{a}_t). \tag{5}$$

We quantify epistemic uncertainty as ensemble disagreement by computing the variance over the ensemble predictions averaged over latent state dimensions $J$ and use it as the new reward term $r_t^{\text{dis}}$:

$$r_t^{\text{dis}} = \frac{1}{J} \sum_{j=1}^{J} \text{Var}(\hat{z}_{j,t}^n), \tag{6}$$

Thus, the model learns to predict two intrinsic rewards $(\hat{r}_t^{\text{sem}}, r_t^{\text{dis}})$ for a state-action-pair (Fig. 2b).

## 2.3 Exploration policy: Go and Explore with SENSEI

We could use a weighted sum of the two intrinsic reward signals, e.g. $r_t^{\text{sem}} + \beta r_t^{\text{dis}}$, as the overall reward $r_t^{\text{expl}}$ for optimizing an exploration policy. However, ideally the weighting of the two signals should dynamically depend on the situation. In uninteresting states we want the agent to mostly focus on optimizing interestingness (via $r_t^{\text{sem}}$). However, once the agent has found an interesting state, we would like the agent to branch out and discover new behavior (via $r_t^{\text{dis}}$). This follows the principle of Go-Explore (Ecoffet et al., 2021), where the agent should first **go** to an important subgoal and **explore** from there. We implement this using an adaptive threshold parameter $\beta \in \{\beta^{\text{go}}, \beta^{\text{explore}}\}$, where $\beta^{\text{explore}} > \beta^{\text{go}}$, whose value depends on the following switching criteria:

$$r_t^{\text{expl}} = \hat{r}_t^{\text{sem}} + \begin{cases} \beta^{\text{explore}} r_t^{\text{dis}}, & \text{if } \hat{r}_t^{\text{sem}} \geq Q_k(\hat{r}^{\text{sem}}); \\ \beta^{\text{go}} r_t^{\text{dis}}, & \text{otherwise.} \end{cases} \tag{7}$$

Here $Q_k$ denotes the $k^{\text{th}}$ quantile of $\hat{r}^{\text{sem}}$, which we estimate through an exponential moving average. Thus, until a certain level of $\hat{r}^{\text{sem}}$ is reached, the exploration reward mainly aims at maximizing interestingness. After exceeding this threshold, exploration more strongly favors uncertainty-maximizing

behaviors. As soon as the agent enters a less interesting state with $\hat{r}^{\text{sem}} < Q_k$, SENSEI switches back to mainly optimizing for semantic interestingness. The two trade-off factors $\beta^{\text{go}}$ and $\beta^{\text{explore}}$, as well as the quantile $k$ are hyperparameters. More details on this adaptation and hyperparameters can be found in Suppl. B. We then learn an exploration policy based on $r_t^{\text{expl}}$ using the DreamerV3 algorithm (Hafner et al., 2023).

## 3 RELATED WORK

**Intrinsic rewards** are applied either to facilitate exploration in tasks where direct rewards are sparse or in a task-agnostic setting where they help collect diverse data. There are many different reward signals that could be useful for efficient exploration of the environment (Baldassarre & Mirolli, 2013), such as prediction error (Schmidhuber, 1991; Pathak et al., 2017; Kim et al., 2020), novelty and Bayesian surprise (Storck et al., 1995; Blaes et al., 2019; Paolo et al., 2021), learning progress (Schmidhuber, 1991; Colas et al., 2019; Blaes et al., 2019), empowerment (Klyubin et al., 2005; Mohamed & Jimenez Rezende, 2015), metrics for state-space coverage (Bellemare et al., 2016; Tang et al., 2017; Burda et al., 2019) and regularity (Sancaktar et al., 2024). While effective for low-dimensional observations, such objectives could be more challenging to apply in the case of high-dimensional image observations. An approach for exploration in image-based environments is to employ low-dimensional goal spaces (Colas et al., 2019; OpenAI et al., 2021; Nair et al., 2018; Pong et al., 2019; Zadaianchuk et al., 2021; Mendonca et al., 2021). An alternative, more sample efficient, direction, is to learn latent world models (Hafner et al., 2019a; 2023; Gumbsch et al., 2024) from visual observations and use these world models for model-based exploration (Pathak et al., 2019; Sekar et al., 2020). In particular, Plan2Explore (Sekar et al., 2020) uses ensemble disagreement of latent space dynamics predictions as an intrinsic reward. While this is a very general strategy for exploration, this could be limited in more challenging environments where semantically meaningful or goal-directed behavior (Spelke, 1990) is needed for efficient exploration.

**Exploration with foundation models:** Recent improvements of in-context learning of LLMs open additional ways to explore using human bias of interestingness during exploration (Klissarov et al., 2023; Du et al., 2023; Zhang et al., 2023a) and skill learning (Colas et al., 2020; 2023; Zhang et al., 2023b). MOTIF (Klissarov et al., 2023) leverages LLMs to generate intrinsic rewards by evaluating pairs of event captions to derive rewards, demonstrating its efficacy in the complex game of NetHack (Küttler et al., 2020). This approach has shown that intrinsic rewards can sometimes outperform direct reward maximization strategies. Similarly, ELLM (Du et al., 2023) uses LLMs to guide RL agents towards goals that are meaningful and useful, based on the agent's current state (represented by text), showing improved task coverage in the Crafter environment (Hafner, 2021). Furthermore, OMNI (Zhang et al., 2023a) introduces a novel method to prioritize tasks by modeling human notions of interestingness using LLMs. Thereby, OMNI enhances the open-ended learning process by focusing on tasks that are not only learnable but also generally interesting. LAMP (Adeniji et al., 2023) proposes to use VLMs for reward modulation in an RL setup. First, a set of potential tasks are generated with an LLM and then LAMP uses VLMs to generate rewards for these tasks to learn a language-conditioned policy in the pretraining phase. This policy is later finetuned with actual task rewards.

**Reward-shaping through VLMs:** Most works that rely on VLMs as reward sources try to solve the reward specification problem in RL. In these works, often a task is assumed to be described as a language caption (Cui et al., 2022; Rocamonde et al., 2023; Baumli et al., 2023; Adeniji et al., 2023), as a goal image (either in-distribution or out-of-distribution) (Cui et al., 2022), or as a demonstration video of the task (Sontakke et al., 2023). In particular, RL-VLM-F (Wang et al., 2024) uses a very similar setup to ours to generate reward functions. Images of initial rollouts are compared pairwise using a VLM to distill a reward function via MOTIF (Klissarov et al., 2023). However, unlike in our work, the VLM is explicitly prompted with the task, whereas we attempt to distill an environment-specific but general exploration reward. Furthermore, SENSEI assumes a model-based setup to learn a world model instead of optimizing a policy based on the distilled reward function.

## 4 RESULTS

Our experiments set out to empirically evaluate the following questions:

1. Does the distilled reward function $R_\psi$ from VLM annotations encourage interesting behavior?

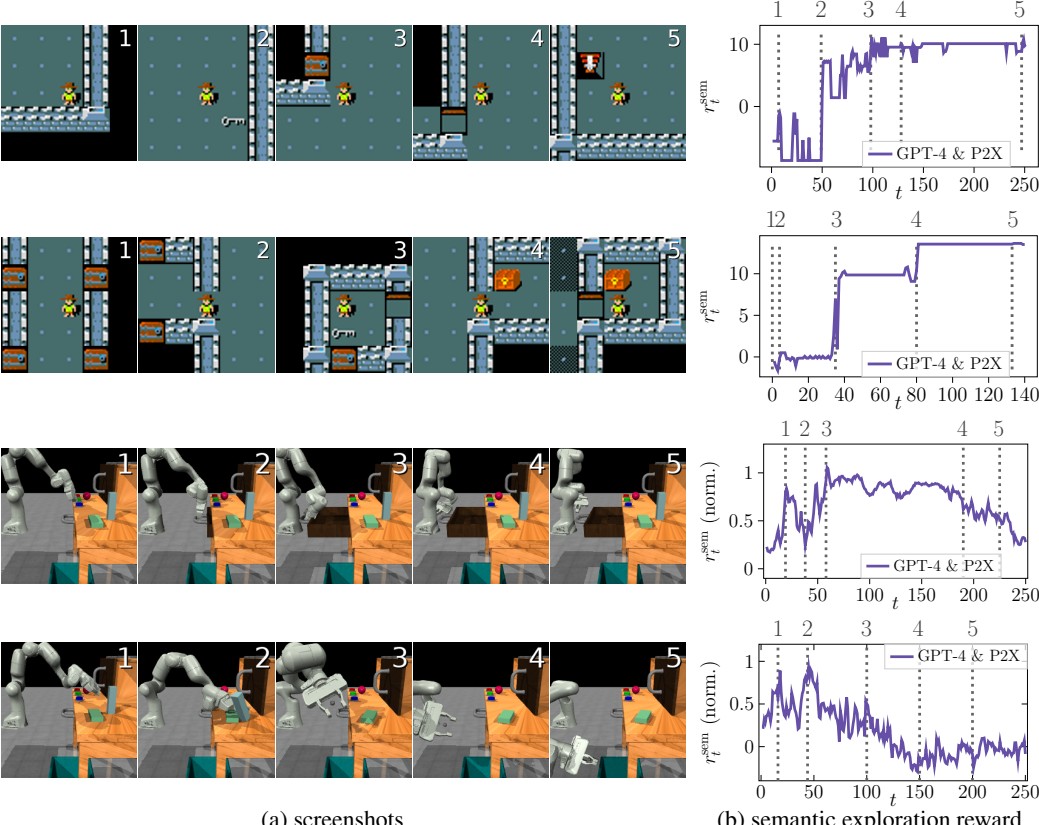

(a) screenshots       (b) semantic exploration reward

Figure 3: **Semantic exploration rewards for example trajectories**: From top to bottom we show example trajectories for Minihack `KeyRoom-S15` and `KeyChest` (see Fig. 8 for map views) and two Robodesk episodes. We showcase rewards from VLM-MOTIF distilled from GPT-4 annotations using Plan2Explore (P2X) data. The reward trajectories peak at the "interesting" moments of exploration, such as opening a drawer in Robodesk or picking up the key in MiniHack.

2. Can SENSEI discover semantically meaningful behavior during task-free exploration?

3. Is the world model learned from exploration suitable for subsequent learning to efficiently solve downstream tasks?

We answer these questions by (1) illustrating the semantic rewards, (2) quantifying the behavior discovered by SENSEI during task-free exploration, and (3) employing the explored world models to subsequent learning of task-based policies. We use two fundamentally different types of environments:

**MiniHack** (Samvelyan et al., 2021) is a sandbox to design RL tasks based on NetHack (Küttler et al., 2020). In MiniHack, an agent needs to navigate dungeons by interacting with its environment in meaningful ways, e.g. apply a key to open a door. We tested two tasks: fetching a key in a huge room to unlock a smaller room with an exit (`KeyRoom-S15`) or fetching a key to open a hidden chest in a maze of rooms (`KeyChest`). MiniHack uses discrete actions. As observations we use pixel-based, egocentric views around the agent and a binary flag indicating key pick-ups (details in Suppl. C.2).

**Robodesk** (Kannan et al., 2021) is a multi-task RL benchmark in which a simulated robotic arm can interact with various objects on a desk, including buttons, two types of blocks, a ball, a sliding cabinet, a drawer, and a bin. For different objects, there exist different tasks, e.g. `open_drawer` or `push_flat_block_in_bin`, with individual sparse rewards. Robodesk uses pixel-based observations and continuous actions controlling the end-effector (more details in Suppl. C.1).

In both environments, we compare SENSEI to Plan2Explore (Sekar et al., 2020), the current state-of-the art in model-based exploration with pixel-based observations. We collect the initial exploration dataset $\mathcal{D}^{\text{init}}$ of SENSEI with 500k steps of Plan2Explore exploration for MiniHack and 1M steps for Robodesk. For data annotation, we use GPT-4 (details in Suppl. C.3).

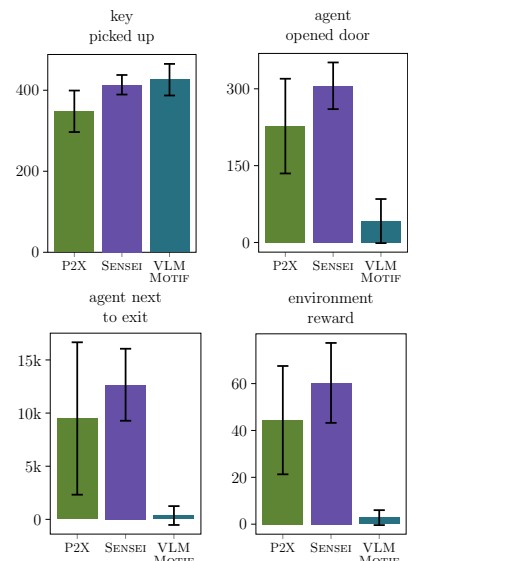
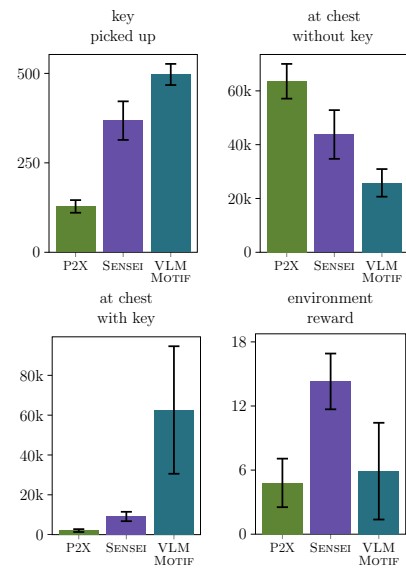

(a) interactions and rewards in `KeyRoom-S15`        (b) interactions and rewards in `KeyChest`

Figure 4: **Interactions in MiniHack**: We plot the mean number of interactions with task-relevant objects and the environment reward (unknown to the agents) collected by SENSEI, Plan2Explore (P2X) and pure VLM-MOTIF (SENSEI with no information gain, i.e. $\beta = 0$) for `KeyRoom-S15` (**a**) and `KeyChest` (**b**). Error bars show the standard deviation (10 seeds).

## 4.1 REWARD FUNCTION OF SENSEI

We illustrate how the distilled VLM-MOTIF reward function $R_\psi$ assigns semantic exploration rewards $r_t^{\text{sem}}$ for exemplary sequences from MiniHack and Robodesk environments, as shown in Fig. 3. In the Minihack environments, we clearly see jumps in reward $r_t^{\text{sem}}$ for significant events. Frames 2 & 3 in `KeyRoom-S15` and `KeyChest` respectively, are right before the key is picked up. Later, $r_t^{\text{sem}}$ increases further once the agent is at the door or chest with a key (Frame 3 in `KeyRoom-S15` and Frames 4&5 in `KeyChest`). For Robodesk, we see that as the robot is interacting with objects, $r_t^{\text{sem}}$ also increases for the examples of opening the drawer and interacting with the blocks.

## 4.2 TASK-FREE EXPLORATION

### 4.2.1 MINIHACK

We quantify the interactions uncovered by SENSEI during task-free exploration in two tasks of MiniHack. For task-relevant events, the mean number of interactions are plotted in Fig. 4. SENSEI focuses more on semantically interesting interactions compared to Plan2Explore, e.g. picking up a key, opening a locked door, or finding the chest with a key. As a result, SENSEI completes both tasks more frequently than Plan2Explore during task-free exploration, as evident by the higher number of collected rewards. We believe this indicates that SENSEI is well suited for initial task-free exploration in these environments, enabling the discovery of state-space regions crucial for solving downstream tasks encoded by human-designed sparse rewards.

**Is information gain crucial for SENSEI?** We showcase results for exploration with pure semantic reward $r_t^{\text{sem}}$, corresponding to SENSEI without information gain reward $r_t^{\text{dis}}$ ($\beta = 0$). In this ablation, we emphasize the crucial role of the information gain objective. Optimizing only for the semantic reward $r_t^{\text{sem}}$ can cause the agent to get stuck in local optima and hinder its exploration around those optima. Let's for instance look at the KeyRoom task: the agent with pure VLM-MOTIF rewards picks up the key many times throughout exploration as this also corresponds to a fairly high semantic reward. However, it fails to explore the room well enough after picking up the key to find and open the door and get to the exit, as reflected in the interaction metrics shown in Fig. 4. We observe a similar

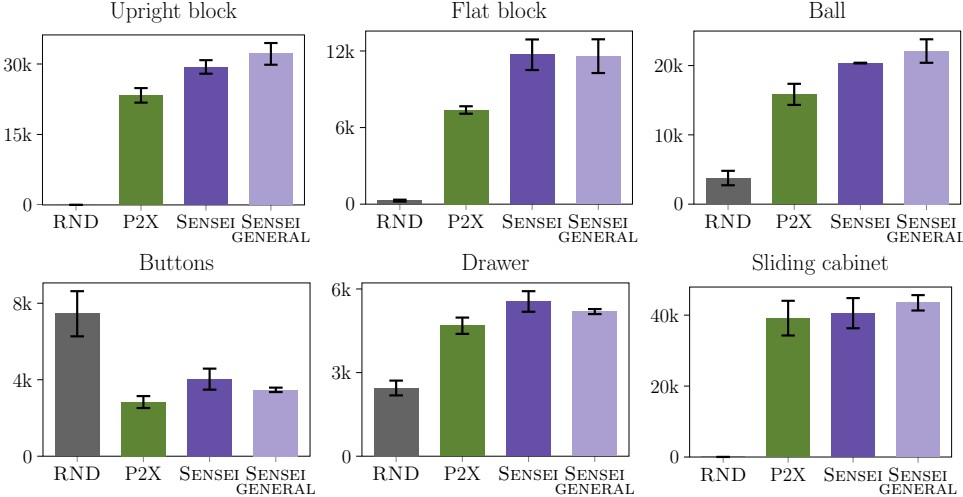

Figure 5: **Interactions in Robodesk**: We plot the mean over the number of interactions with any object during 1M steps of exploration for SENSEI (with an environment description provided by us), a more general variant of SENSEI with a VLM-generated environment description (SENSEI GENERAL), Plan2Explore (P2X), and Random Network Distillation (RND). Error bars show the standard deviation (3 seeds).

scenario for KeyChest: although the pure VLM-MOTIF agent gets to the chest after having picked up the key fairly often, it collects significantly less rewards compared to SENSEI. For the episode to end, the agent needs to use the key to open the chest. The pure VLM-MOTIF agent, however, simply hovers around the chest. As being at the chest with a key is an "interesting" state and opening a chest immediately terminates the episode, there is no real incentive for the agent to explore what would happen if the chest was opened. This ablation shows the importance of combining novelty and usefulness in order to continually push the frontier of experience.

### 4.2.2 ROBODESK

Next, we analyze exploration in the challenging visual control suite of Robodesk. Here we compare 1M steps of exploration in SENSEI with Plan2Explore and Random Network Distillation (RND, Burda et al., 2019), a strong model-free exploration approach that uses prediction errors of random image embeddings as intrinsic rewards to maximize state space coverage. For Robodesk, in order to deal with occlusions, we use images from two camera angles for GPT annotations. We only keep the GPT annotation if the ranking for both agree. The world model and the distilled VLM-MOTIF network use only the right camera image, as in Plan2Explore or RND (see Suppl. C.3.1 for more details). Fig. 5 plots the mean number of object interactions during exploration for the two methods. On average, SENSEI interacts more with most available objects than the baselines. As a result, in a majority of tasks SENSEI receives more task rewards during exploration than Plan2Explore or RND (shown in Suppl. D.3). Qualitatively, we observe that Plan2Explore mostly performs arm stretches[3], whereas RND mostly moves the arm around in the center of the screen, occasionally hitting objects or buttons. Thus, our semantic exploration reward seems to lead to more meaningful behavior than pure epistemic uncertainty-based exploration, even in a low-level motor control robotic environment.

**Is an environment description by a human expert necessary for SENSEI?** We investigate whether SENSEI relies on the external environment description provided by us, and compare against a version of SENSEI using a more general prompting strategy (SENSEI GENERAL). During data annotation, SENSEI GENERAL first prompts the VLM for an environment description given an image

---

[3]Interestingly, this can still lead to solving tasks during exploration. For example, stretching the arm against the sliding cabinet can close it, and stretching the arm toward the upright block can push it off the table.

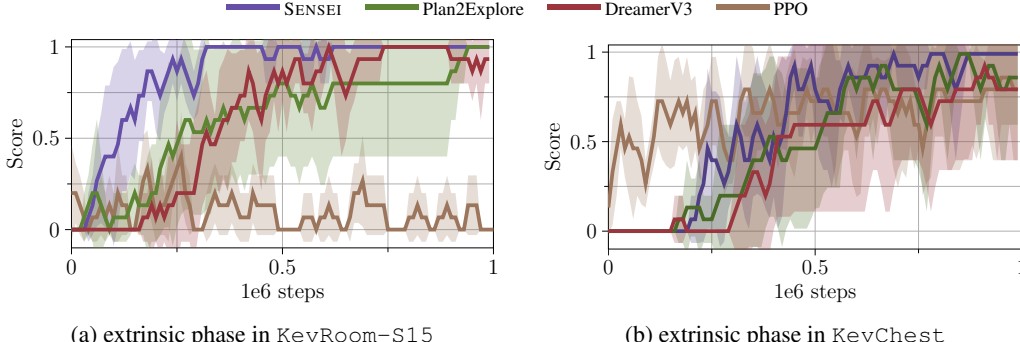

(a) extrinsic phase in `KeyRoom-S15`          (b) extrinsic phase in `KeyChest`

Figure 6: **Downstream task performance in MiniHack**: We plot the mean of the episode score obtained during evaluation for the MiniHack tasks (**a**) `KeyRoom-S15` and (**b**) `KeyChest`, with world models learned from SENSEI vs. Plan2Explore (P2X) exploration. We also show results for learning a task policy from scratch with DreamerV3, and the model-free baseline PPO. Shaded areas depict the standard deviation (5 seeds) and we apply smoothing over the score trajectories with window size 3.

of the simulation and uses this context to annotate the dataset of preferences (details in Suppl. C.3.2). As shown in Fig. 5, SENSEI GENERAL interacts roughly as often with the relevant objects as SENSEI, outperforming both exploration baselines Plan2Explore and RND in terms of overall number of object interactions. Thus, injecting external expert knowledge about the environment to the prompts is not needed to train SENSEI and this step can be fully automated by a VLM. This further cements the generality of our approach.

**Ablations**   We perform ablations to see 1) how noisy annotations from VLMs affect SENSEI in comparison to an oracle annotator and 2) how much the behavior richness of the initial dataset can help bootstrap SENSEI performance in Fig. 11. We observe that as VLMs get better and better, there is indeed more to gain from SENSEI, and richer exploration data helps SENSEI bootstrap faster. See Suppl. D.2 for more information.

### 4.3   FAST DOWNSTREAM TASK LEARNING

We hypothesize that world models learned from richer exploration would enable model-based RL agents to quickly learn to solve new downstream tasks. We investigate this in Minihack by running DreamerV3 (Hafner et al., 2023) using the previously explored world models to learn a novel task-based policy. To this end, we jump-start policy training by initializing DreamerV3 with the pre-trained world models from our initial 500K steps of exploration (see Sec. 4.2). We compare world models from task-free exploration with either SENSEI or Plan2Explore. Additionally, we compare running DreamerV3 and training Proximal Policy Optimization (PPO, Schulman et al. 2017), a state-of-the-art model-free baseline, from scratch.

Figure 6 shows the performance of task-based policies over training. A previously explored world model from SENSEI allows the agent to learn to solve the task faster than all other baselines. As SENSEI allocates more resources to explore the relevant dynamics in the environment compared to Plan2Explore, e.g. opening the chest more frequently instead of just being near the chest, a more useful world model is learned that later aids policy optimization. In addition, the world models explored with Plan2Explore do not capture the task-relevant dynamics of the environment as accurately, which slows down policy learning compared to SENSEI. Compared to learning a task policy from scratch with DreamerV3, we do not observe a consistent trend of improvement from task-free exploration with Plan2Explore. The model-free baseline PPO shows the first successes in the tasks early during training but on average takes longer to learn to reliably solve the task. In `KeyRoom`, for example, PPO takes more than 20M steps to consistently solve the task across all random seeds (full PPO curves in Supp. Fig. 10). Thus, in this task SENSEI outperforms PPO by roughly two orders of magnitude. This shows the improved sample efficiency of our approach, combining foundation model-guided exploration and model-based RL.

## 5 DISCUSSION

We have introduced SENSEI, a framework for guiding the intrinsically motivated exploration of model-based agents through foundation models without assuming access to expert data, high-level actions, or perfect environment captions. SENSEI bootstraps its model of interestingsness from previously generated play data with e.g. information gain. On this dataset, SENSEI prompts a VLM to compare images with respect to their interestingness and distills a reward function for semantically grounded exploration. SENSEI learns an exploration policy via model-based RL using two sources of intrinsic rewards: (1) trying to reach states with high semantic interestingness and (2) branching out from these states to maximize epistemic uncertainty. We show that in a simulated robotic environment, this strategy leads to more object manipulations than pure information gain-oriented exploration. Similarly, we demonstrate that SENSEI discovers more semantically meaningful interactions, such as applying a key to open a chest, when exploring the video game-like environments of MiniHack. In both environments, SENSEI accumulates more rewards than the state-of-the-art exploration method Plan2Explore (Sekar et al., 2020), by "accidentally" solving human-designed sparse reward tasks already during exploration.

**Internal model of interestingness**   In contrast to other methods of foundation model-guided exploration (Klissarov et al., 2023; Wang et al., 2024), SENSEI combines semantic exploration with world model learning, thereby learning an internal model of "interestingness." This is a sensible design choice when working with world models (as detailed in Suppl. A.3). Previous variants of MOTIF (Klissarov et al., 2023; Wang et al., 2024) require the complete observation in order to compute the foundation model-based reward. After training, SENSEI can predict semantic rewards from its latent states, thus, also evaluating imagined and hypothetical states. As demonstrated by our experiments, this can lead to significantly faster policy learning, since both VLM guidance as well as model-based RL improve sample efficiency.

**Future work**   We hypothesize that SENSEI learns a versatile world model from exploration, as it is trained on a more interaction-rich and interesting dataset which could enable the agent to quickly learn to solve downstream tasks. We showcase this in the MiniHack tasks `KeyRoom-S15` and `KeyChest`. In future work, we plan to examine this for zero-shot model-based planning (Sancaktar et al., 2022). Furthermore, in Suppl. D.2, by comparing two sources for the initial play dataset $\mathcal{D}^{\text{init}}$, we observe that SENSEI reinforces the trends existing in the initial exploration round, while still seeing improvements across different types of interactions. To further reinforce more complex types of interactions, our reward function could be refined in a new round of VLM annotations from a SENSEI run. As a result, SENSEI could potentially unlock increasingly complex sequences of behavior with each generation. We believe SENSEI has the potential to be applied to realistic simulations or real-world tasks. Photorealism of observations are likely to help VLM annotations because a large portion of their training data comes from real world photos or videos. Thus, we think that SENSEI could scale well to more realistic applications.

**Limitations**   SENSEI benefits from fully-observable observations, e.g. images that capture all relevant aspects of the environment. The VLM annotations, and as a result the distilled reward function, degrade when dealing with occlusions and the lack of depth information from a single image in visually-complex environments such as Robodesk. In this work, we used a second camera angle for VLM annotations in Robodesk to reduce noise. We believe this can be remedied further in future work by using videos for VLM annotations to better convey temporal or partially observable information.

### REPRODUCIBILITY STATEMENT

We provide all details and hyperparameters of our proposed method, as well as the baselines in the supplementary (see Suppl. A and Suppl. B). We will make the code, annotated datasets, distilled VLM-MOTIF networks and world model checkpoints publicly available upon acceptance, such that all results in the paper can be reproduced.

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

# Supplementary Material for:
# SENSEI: Semantic Exploration Guided by Foundation Models to Learn Versatile World Models

## A SENSEI: IMPLEMENTATION DETAILS

### A.1 WORLD MODEL

**RSSM** We base our RSSM implementation on DreamerV3 (Hafner et al., 2023). For MiniHack we use the small model size setting with roughly 18M parameters ($h_t$ dimensions: 512, CNN multiplier: 32, dense hidden units: 512, MLP layers: 2). For the more complicated Robodesk environment, we use the medium model size with around 37M parameters ($h_t$ dimensions: 1024, CNN multiplier: 48, dense hidden units: 640, MLP layers: 3). By default, when the input observation $o_t$ is only an image, it is en- and decoded through CNNs. For MiniHack, we have an additional inventory flag that is processed by a separate MLP, as is customary for the Dreamer line of work when dealing with multimodal inputs (Wu et al., 2023). The MLP decoder outputs a Bernoulli distribution from which we sample the decoded inventory flag.

**Reward predictors** To handle rewards of widely varying magnitudes, DreamerV3 uses twohot codes predicted in symlog space when predicting rewards (Hafner et al., 2023). We use the same setup for all reward prediction heads, i.e., for extrinsic rewards $r_t^i$ for task $i$ or the semantic exploration reward $r_t^{\text{sem}}$. During task-free exploration, the gradients from reward predictions are stopped to not further affect world model training. We do this to keep the world model somewhat task-agnostic to later reuse it for multiple tasks. Similarly, to avoid overfitting to the exploration regime, we also stop the gradients from the semantic reward prediction heads.

**Plan2Explore** Both our Plan2Explore baseline as well as our ensemble predictors (Eq. 5) are based on the re-implementation on top of DreamerV3. The most notable difference is that in original Plan2Explore the ensemble is trained to predict image encodings (Sekar et al., 2020), whereas the new version is trained to predict stochastic states $z_t$. Recent re-implementations (Hafner, 2021; Hafner et al., 2022; Gumbsch et al., 2024) also used Plan2Explore with ensemble disagreement over $z_t$ as a baseline and verified a strong exploration performance.

**Quantile estimation** We update our estimate of the quantile $Q_k(\hat{r}^{\text{sem}})$ whenever we train the exploration policy. For this, we compute the $k$-th quantile of $\hat{r}_t^{\text{sem}}$ in each training batch ($16 \times 16$). We keep an exponential moving average over these estimates with a smoothing factor of $\alpha = 0.99$.

**Reward weighting** In practice, we compute exploration rewards (Eq. 7) using two reward factors for each loss term

$$r_t^{\text{expl}} = \begin{cases} \alpha^{\text{explore}}\hat{r}_t^{\text{sem}} + \beta^{\text{explore}}r_t^{\text{dis}}, & \text{if} \quad \hat{r}_t^{\text{sem}} \geq Q_k(\hat{r}^{\text{sem}}); \\ \alpha^{\text{go}}\hat{r}_t^{\text{sem}} + \beta^{\text{go}}r_t^{\text{dis}}, & \text{otherwise.} \end{cases} \tag{8}$$

i.e. $\alpha$ to scale $\hat{r}_t^{\text{sem}}$ and $\beta$ to scale $r_t^{\text{dis}}$. When training the value function with DreamerV3, the scale of the reward sources are normalized. To compute this normalization for the exploration policy we use $\alpha^{\text{explore}}$ and $\beta^{\text{explore}}$ of the high percentile region of interestingness ($\geq Q_k$).

### A.2 SEMANTIC REWARD DISTILLATION: VLM-MOTIF

For the semantic reward function $R_\psi : \mathcal{O} \to \mathbb{R}$, we use a 2D-convolutional neural network to encode the images. We use 3 convolutional layers, where we progressively increase the number of channels to `num_channels_max` $= 64$. The output then gets downsampled via max pooling before going into a two-layer MLP with hidden dimensions 256 & 512 and outputting the scalar reward value. Additionally, in MiniHack we include inventory information via a separate multi-layer perceptron (MLP) head, consisting of 2 layers with 512 hidden units. The extracted features are concatenated with the image features and get further processed by the output MLP. The training hyperparameters for all $R_\psi$ can be found in Suppl. B.

### A.3 Design Choice: Semantic Reward Predictions

World models typically encode and predict dynamics fully in a self-learned latent state (Ha & Schmidhuber, 2018; Hafner et al., 2023; Hansen et al., 2024). Thus, for a world model to predict $r_t^{\text{sem}}$ at any point in time $t$, we need a mapping from latent states to semantic rewards. We chose to directly predict $\hat{r}_t^{\text{sem}}$ using a reward prediction head of the RSSM. Another option would be to decode the latent state to images and use those as inputs for MOTIF. However, we believe this has several disadvantages: 1) Decoding latent states to images is a computationally costly step that would significantly decrease our SENSEI's computational efficiency. 2) We would use an indirect target (the image) instead of the direct target ($\hat{r}_t^{\text{sem}}$) for training the semantic reward predictions. There would be no gradient to correct somewhat reasonable image predictions that lead to inconsistent reward predictions at a given state. 3) The image predictions of the RSSM can contain artifacts, blurriness or hallucinations. Since MOTIF is only trained on real images from the simulation, we will likely encounter out-of-distribution errors.

## B Hyperparameters

We provide the hyperparameters used for the world model, exploration policy, VLM-MOTIF annotations & reward model training as well as the environment-specific settings.

| Name | Value | | |
|---|---|---|---|
| | Robodesk | KeyRoom | KeyChest |
| **World Model** | | | |
| RSSM size | M | S | S |
| Ensemble size $N$ | 8 | 8 | 8 |
| Train ratio | 512 | 512 | 512 |
| **Exploration policy** | | | |
| Quantile | 0.75 - 0.85 -0.75 - 0.80 | 0.90 | 0.90 |
| $\alpha^{\text{explore}}$ | 0.1 - 0.1 - 0.05 - 0.01 | 0.3 | 0.25 |
| $\beta^{\text{explore}}$ | 1 - 1 - 1 - 1 | 1 | 1 |
| $\alpha^{\text{go}}$ | 1 - 1 - 1 - 1 | 1 | 1 |
| $\beta^{\text{go}}$ | 0 - 0 - 0 - 0 | 0.1 | 0.05 |
| **Annotations for MOTIF** | | | |
| VLM | GPT-4 turbo (right) & GPT-4 omni (left) | GPT-4 omni | GPT-4 omni |
| Temperature | 0.2 | 0.2 | 0.2 |
| Dataset size | 200K | 100K | 100K |
| Image res. | $224 \times 224$ | $80 \times 80$ | $80 \times 80$ |
| **MOTIF Training** | | | |
| Batch size | 32 - 64 - 32 - 32 | 32 | 32 |
| Learning rate | $10^{-5}$ - $10^{-5}$ - $3 \times 10^{-5}$ - $3 \times 10^{-5}$ | $10^{-4}$ | $10^{-4}$ |
| Weight decay | $10^{-5}$ - 0 - 0 - 0 | $10^{-5}$ | $10^{-4}$ |
| **Environment** | | | |
| Action repeat | 2 | 1 | 1 |
| Episode length | 250 | 600 | 800 |
| Steps of exploration | 1M | 500K | 500k |

For the exploration policy in Robodesk we use different values for the four different variants tested. The values listed here stand for, from left to right: GPT-4 with Plan2Explore data using two camera angles for VLM annotations, GPT-4 with Plan2Explore data using only the right camera angle for annotations, Oracle with Plan2Explore data, and Oracle with CEE-US data (corresponding to a more interaction-rich exploration dataset for $\mathcal{D}^{\text{init}}$). The VLM-MOTIF training hyperparameters are also listed in the same order.

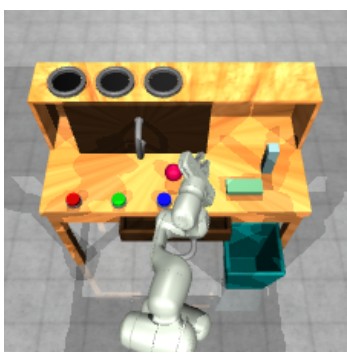
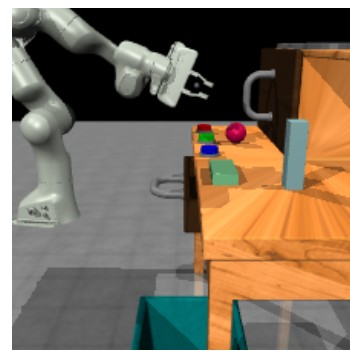

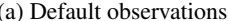

(a) Default observations           (b) Our observations

Figure 7: **Robodesk environment.** We modify the default top-down camera view **(a)** to a side view with less occlusion **(b)**.

**Image resolution**   For the world model we use $64 \times 64$ images for all environments. However, for the GPT annotations we use higher resolution images, as shown in the table. Inside the environment `step` function, the rendering is performed at these higher resolutions, and this image is input to the semantic reward function $R_\psi$. The image is then scaled down to $64 \times 64$ as part of the observation that the RSSM is trained on.

**Baselines**   We run DreamerV3 with the same world model setup as SENSEI and Plan2Explore. We use an open source PPO (Schulman et al., 2017) implementation of Hafner (2024)[4] optimized to work well across multiple environments with a fixed set of hyperparameters (details in Hafner et al., 2023, supplementary material). We build our RND (Burda et al., 2019) implementation on top of PPO. For the predictor and target network we use a ResNet with 3 convolutional layers followed by 5 dense layers. We only use the intrinsic reward to train a PPO agent. Intrinsic rewards are normalized as outlined in Burda et al. (2019). While Burda et al. (2019) also normalize input observations through a running statistics, we found that using LayerNorm at the input layer leads to slightly more interactions in Robodesk.

## C   ENVIRONMENT DETAILS

### C.1   ROBODESK

Robodesk (Kannan et al., 2021) is a multi-task RL benchmark in which a robot can interact with various objects on a desk. We use an episode length of 250 time steps.

**Observations**   Robodesk uses only an image observation, depicting the current scene, which we scale down ($64 \times 64$ pixels). However, we found that the default top-down view often had occlusions and was hard to interpret from a single image (Fig. 7a). Thus, we used a different camera angle showing the robot from one side (Fig. 7b). With this view objects and the drawer were rarely occluded; however, lights that turn on from button presses were not as visible anymore.

**Actions**   The continuous 5-dimensional actions control the movement of the end effector. We use an action repeat of 2 to speed up the simulation. Thus, 1M steps of exploration correspond to 2M actions in the environment.

**Interaction metrics**   We track how often the robot interacted with different objects to quantify the behavior during exploration by tracking the velocity of joints and object positions. For buttons, sliding cabinet, or drawer, we check if the joint position changes more than a fixed value (0.02). For all other objects, we check if any of their $x$-$y$-$z$ velocities exceed a threshold (0.02).

---

[4]https://github.com/danijar/embodied, version v1.2

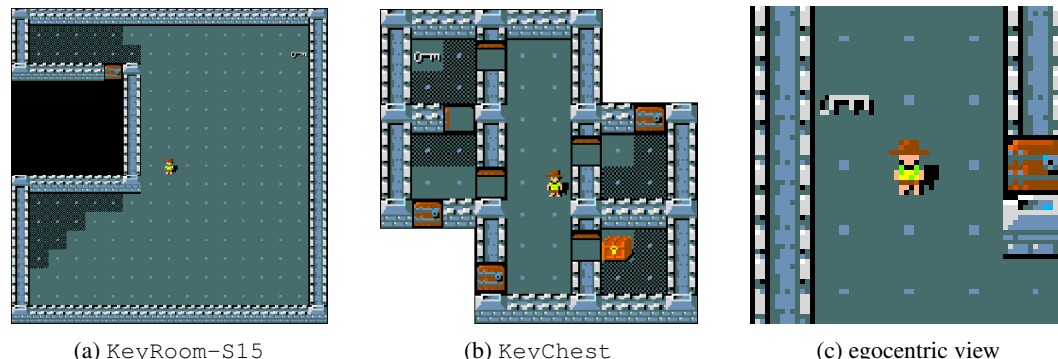

(a) `KeyRoom-S15`          (b) `KeyChest`          (c) egocentric view

Figure 8: **MiniHack** : We consider two tasks `KeyRoom-S15` **(a)** and `KeyChest` **(b)**. The agent receives an egocentric view of the environment as its observation **(c)**.

**Tasks** We use the sparse reward versions of all the tasks available in the environment. For some tasks, we add easier versions. All tasks describe interactions with one or multiple objects:

- **Buttons**: Pushing the red (`push_red`), blue (`push_blue`), or green (`push_green`) button.
- **Sliding cabinet**: Opening the sliding cabinet fully (`open_slide`).
- **Drawer**: Opening the drawer fully or opening it slightly (`open_drawer_light`). We introduced the latter task.
- **Upright Block**: Lifting the upright block (`lift_upright_block`), pushing it off the table (`upright_block_off_table`) or putting it into the shelf (`upright_block_in shelf`).
- **Flat Block**: Lifting the flat block (`lift_flat_block`), pushing it off the table (`flat_block_off_table`), into the bin (`flat_block_in_bin`), or into the shelf (`flat_block_in_shelf`).
- **Both blocks**: Stacking both blocks (`stack`).
- **Ball**: Lifting the ball (`lift_ball`), dropping it into the bin (`ball_in_bin`) or putting it into the shelf (`ball_in_shelf`).

### C.2 MINIHACK

**Observations** In MiniHack multiple observation and action spaces are possible. We use egocentric, pixel-based observations centered on the agent ($\pm 2$ grids, example in Fig. 8c). In addition to that, we provide the agent's inventory. By default, in MiniHack the inventory is given as an array of strings (UTF8 encoded), and different player characters have different starting equipment based on the character classes of NetHack. We simplify this by providing only a binary flag that indicates if the agent has picked up a new item. This is sufficient for the problems we consider, in which maximally one new item can be collected and starting equipment cannot be used.

**Environments** Here we detail the environments we tackle:

In the benchmark `KeyRoom-S15` problem (Fig. 8a), the agent needs to fetch a key in a large room ($15 \times 15$ grids) to enter a smaller room and find a staircase to exit the dungeon. We use the default action space but enable autopickup and therefore remove the `PICKUP` action. We use an episode length of 600 time steps, which is 1.5 times longer than the default episode length.

`KeyChest` is a novel environment designed by us, based on `KeyCorridorS4R3` from MiniGrid (Chevalier-Boisvert et al., 2024) (see Fig. 8b). The agent starts in a corridor randomly connected to different rooms. A key is hidden in one room and a chest in another room. The goal is to open the chest with the key in the inventory. Object positions are randomized. The action space for this task contains 5 discrete actions for moving the agent in 4 cardinal directions (`UP`, `RIGHT`, `DOWN`, `LEFT`) and an `OPEN`-action to open a chest when standing next to it with a key in the inventory. We enable auto-pickup, so no additional action is needed to pick up the key when stepping on it. We use an episode length of 800 time steps.

**Rewards**   All environments use a sparse reward of $r_t = 1$, which the agent only receives upon accomplishing the task. A small punishment ($r_t = -0.01$) is given, when the agent performs an action that does not alter the screen.

**Image remapping**   Empirically, we found that GPT-4 may encounter problems if we provide the image observations as is. For example, when using the default character in the `KeyRoom-S15` environment (Rogue), GPT-4 sometimes throws content violation errors. We suspect that this is due to the character wearing a helmet with horns, which could be mistaken for demonic or satanic imagery. Thus, we pre-processed the images before returning them from the environment. We render all characters as the Tourists, a friendly looking character with a Hawaiian shirt and straw hat. Furthermore, GPT-4 sometimes mistakes entrance staircases for exit staircases. Since the entrance staircases serve no particular purpose and are not different from the regular floor, we remap all entrance staircases to floors.

### C.3   VLM PROMPTING

We prompt the VLM with somewhat general descriptions of the environments that we consider. Here we provide the full prompts for all environments.

#### C.3.1   ROBODESK

In Robodesk, for each query, we provide two observation images (resolution $224 \times 224$) with the following prompt:

```
Here are two images in a simulated environment with a robot in
front of a desk.  Your task is to pick between these images
based on how interesting they are.  Which image is more
interesting in terms of the showcased behavior?  For context
following points would constitute interestingness:  (1) The
robot is currently holding an object in its gripper.  (2)
The robot is pushing an object around or pushing a button or
opening the drawer or interacting with entities on the desk.
(3) Objects on the desk are in an interesting configuration:
e.g.  a stack.  Being far away from the desk with the robot
arm retracted or just stretching your arm without interactions,
is a sign the image is not interesting.  Answer in maximum one
word:  0 for image 1, 1 for image 2, 2 for both images and 3 if
you have no clue.
```

Due to occlusions, we annotate the same pair from the initial dataset $\mathcal{D}^{\text{init}}$ with the same prompt using images from two camera angles: right (Fig. 7b) and left (Fig. 9). A pair is deemed valid only if the GPT-4 response is the same across both camera angles, otherwise the pair and the annotation are removed from the dataset. The VLM-MOTIF training as well as the world model training are still executed using only the right camera image, such that during free play with SENSEI we only rely on the right camera images as input.

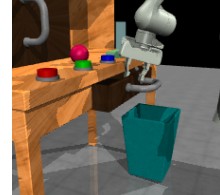

Figure 9: Left camera observation in Robodesk that is used only during the annotation stage with GPT-4.

In Robodesk, we use GPT-4 Turbo (`gpt-4-turbo-2024-04-09`) for the right camera annotations and GPT-4 omni (`gpt-4o-2024-05-13`) for the left camera annotations. To cap the dataset size of pairs at 200K and still make sure that we got contrastive pairs of interesting vs. uninteresting behaviors, we pre-filtered the pairs using our Oracle of interestingness for the Plan2Explore dataset. We believe that in future work this step can be omitted by simply generating more random pairs and expanding the dataset size. After the two-angle image annotation, we use the subset of the dataset where both left and right camera annotations agree. This corresponds to 139409 pairs from the original 200K dataset, or 69.7% of it.

### C.3.2 ROBODESK: ZERO-KNOWLEDGE GENERAL PROMPT

To demonstrate the generality of SENSEI, we also test a version with a general prompt (SENSEI GENERAL), using a multi-turn strategy for getting the annotations for a pair of images. This version of the prompt does not contain any external environment specific knowledge. First, we show a picture from the robotic environment, and ask the VLM (gpt-4o-2024-05-13) to describe what it sees in the scene using the following prompt:

```
I will show you an image from a simulated environment.
Describe the environment to me and what you deem the agent
and then explain what types of behavior / observations would
be useful and interesting for the agent to explore in this
setting?
```

We obtain the following response from GPT:

```
The image depicts a simulated environment featuring a robotic
arm interacting with a table that has various objects on it.
The objects include buttons of different colors, a red ball, a
green cube, and a blue rectangular block.  The setting appears
to be designed for testing robotic manipulation and interaction
with objects.
Agent:  The robotic arm is the agent in this environment.  It
is likely programmed to interact with the objects on the table,
possibly for tasks involving sorting, stacking, or pressing
buttons.
Useful and Interesting Behaviors/Observations:        1.  Object
Manipulation:  - Grasping and moving objects like the
ball, cube, and block.  - Testing different grips and force
applications to handle objects of various shapes and weights.
2.  Button Interaction:  - Pressing buttons to observe effects,
such as changes in the environment or triggering specific
actions.  - Learning which buttons correspond to which
actions or outcomes.  3.  Spatial Awareness:  - Navigating
the arm around the table without knocking over objects.  -
Understanding the spatial relationship between objects for
tasks like stacking.  4.  Task Learning:  - Completing specific
tasks like sorting objects by color or shape.  - Learning
sequences of actions to achieve complex goals.  5.  Feedback
and Adaptation:  - Using sensory feedback to adjust actions
in real-time.  - Adapting to changes in the environment,
such as objects being moved or removed.  6.  Exploration and
Experimentation:  - Experimenting with different strategies
for interacting with objects.  - Exploring the effects of
different actions on the environment.  These behaviors would
help the agent develop skills in manipulation, decision-making,
and adaptive learning, which are crucial for more complex
real-world applications.
```

Next, we show the pair of images that we would like GPT-4 to annotate with the following prompt:

```
Now I will show you two images.  Based on your answer, choose
which one of these images is more interesting in terms of the
showcased behavior?  Answer in maximum one word:  0 for image
1, 1 for image 2, 2 for both images and 3 if you have no clue.
```

We obtain the final annotation for the pair as the answer to this last prompt.

In this experiment, the environment description is generated using the first prompt only once. For subsequent annotations, the initial question-answer pair is included in GPT-4's context window as history. In the second turn, the new image pair is presented using the final prompt. This approach is more cost-efficient, as in-context tokens are less expensive than generated ones, and time-efficient, as generating the full environment description sequentially requires more processing time. With this multi-turn strategy, annotating each image pair costs approximately \$0.004, compared to \$0.002 per annotation with the single-turn prompt.

### C.3.3 MINIHACK

For MiniHack we provide two observation images (resolution $80 \times 80$) and the inventory. For the inventory, we only consider items that were picked up and not items in the agent's starting equipment[5]. We provide the inventory as text descriptions. The different options are shown in purple.

```
Your task is to help play the video game MiniHack.  MiniHack is
a roguelike game where an agent needs to navigate through rooms
and escape a dungeon.  For succeeding in the game, finding
items, collecting items and exploring new rooms is crucial.
Images are egocentric around the agent, who is standing on
a dotted blue floor.  Your task is to pick between two game
states, composed of images and an inventory descriptions, based
on how interesting and useful they are.
Is there any difference between the first and second game
state in terms of how interesting it is?  The images depict
the current view.  {The first agent has a key named The Master
Key of Thievery in their inventory., The second agent has a
key named The Master Key of Thievery in their inventory., Both
agents have a key named The Master Key of Thievery in their
inventory., Both agents have no items in their inventory.},
Think it through and then answer in maximum one word:  0 if the
first state is more interesting, 1 if the second state is more
interesting, 2 if both states are interesting and 3 if nothing
is interesting or you are very unsure.
```

For MiniHack we use GPT-4 omni (`gpt-4o-2024-05-13`).

### C.4 ORACLE FOR INTERESTINGNESS

In Robodesk, we also use an Oracle of interestingness to annotate the pairs as an ablation (see Suppl. D.2). Our goal here is to showcase an upper-bound of performance on SENSEI without the noisiness of VLMs. For the Oracle, we deem a state interesting if: (1) any one of the entities are in motion (here only for the ball we make an exception that the ball should be in motion with the end effector close to it as the ball in the environment is unimpeded by friction), (2) if the drawer is opened, (3) if the drawer/sliding cabinet is not yet in motion, but the end effector is very close to their handles, (4) if the upright and flat blocks are not yet in motion but the end effector is very close to them (almost touching), (5) if the stacking task is solved. With these statements, we essentially cover the range of tasks defined in the Robodesk environment, as they are shown in Fig. 13.

## D EXTENDED RESULTS

### D.1 MINIHACK: EXTENDED RESULTS

Figure 10 shows the full trajectory of evalutation scores for Proximal Policy Optimization (PPO, Schulman et al. 2017) in Minihack when trained until convergence. While PPO manages to learn to

---

[5]The starting equipment is taken from the NetHack game and irrelevant and inaccesible in our tasks.

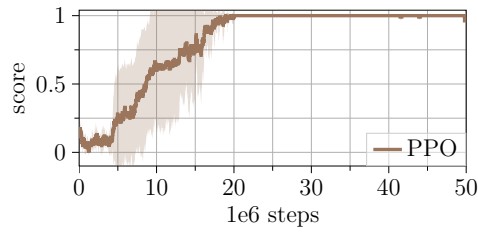
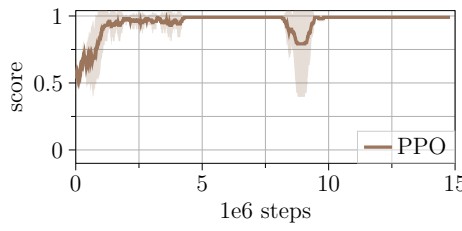

(a) extrinsic phase in `KeyRoom-S15`    (b) extrinsic phase in `KeyChest`

Figure 10: **PPO performance in MiniHack**: We plot the mean episode score obtained by PPO during evaluation for the MiniHack tasks `KeyRoom-S15` (**a**) and `KeyChest` (**b**). Shaded areas depict the standard deviation (5 seeds). We apply smoothing over the score trajectories with window size 20.

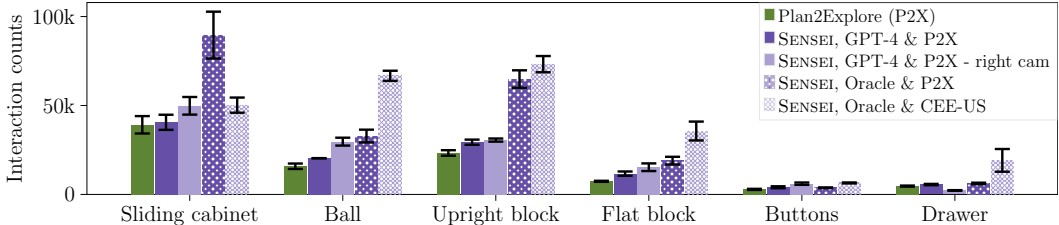

Figure 11: **Interactions in Robodesk**: We plot the mean over the number of interactions with objects in the environment during exploration for different versions of SENSEI (Oracle vs. VLM, CEE-US (Sancaktar et al., 2022) vs. Plan2Explore to create the data to label $\mathcal{D}^{\text{init}}$) and Plan2Explore. We also ablate SENSEI using only the right camera angle for VLM annotations on the Plan2Explore dataset. Error bars show the standard deviation (3 seeds).

solve all tasks, it can be much less sample efficient than the model-based agents we evaluated (see Fig. 6), especially in `KeyRoom-S15`. Here SENSEI outperforms PPO in terms of sample efficiency in one to two orders of magnitude.

## D.2  ROBODESK: SENSEI ABLATIONS

In Robodesk, we compare different versions of SENSEI in order to analyze the effect of the VLM and the initial exploration data on SENSEI performance (Fig. 11). First, we showcase SENSEI results when annotating the initial exploration dataset from Plan2Explore with only the right camera images. In this case, we use the whole 200K pairs in the dataset, without any pruning. In another ablation, we replace the VLM (GPT-4) with a hand-crafted Oracle (see Suppl. C.4 for how the oracle is computed) for annotating the pairs. After the oracle annotations, we distill these preferences into VLM-MOTIF for SENSEI, following the same procedure as before. Furthermore, we compare two initial datasets $\mathcal{D}^{\text{init}}$ of self-supervised exploration collected either by CEE-US (Sancaktar et al., 2022) or by Plan2Explore for the oracle SENSEI versions. CEE-US uses vector-based position of entities for information-gain-based exploration, in comparison to Plan2Explore, which works on the pixel-level. Due to the privileged inputs, $\mathcal{D}^{\text{init}}_{\text{CEE-US}}$ contains more complex interactions. We compare 1M steps of exploration with the four versions of SENSEI and Plan2Explore.

On average, all versions of SENSEI interact more with the objects than Plan2Explore and our semantic exploration reward seems to lead to more object interactions than pure epistemic uncertainty-based exploration. SENSEI with Oracle for both the Plan2Explore and especially the CEE-US initial datasets show the most object interactions. We believe this further showcases that the VLM provides a much noisier signal of interestingness, making it harder to optimize for.

The initial exploration dataset $\mathcal{D}^{\text{init}}$ influences with which objects SENSEI interacts. Qualitatively, we observe Plan2Explore performing mostly arm stretches. Interestingly, this can still lead to solving tasks during exploration. For example, stretching the arm against the sliding cabinet can close it, and stretching the arm toward the upright block can push it off the table. As a result, SENSEI with

Plan2Explore Oracle focuses mainly on the sliding cabinet and the upright block, reinforcing the existing trends in the initial dataset from which VLM-MOTIF is distilled.

For CEE-US data, Oracle SENSEI interacts more with the other objects, such as the ball and the flat block, as well as the drawer. The difference between the Oracle annotator SENSEI versions with CEE-US vs. Plan2Explore data showcases that there is still a lot to be gained from a richer initial dataset for SENSEI, which could be obtained via multiple rounds of SENSEI exploration.

If a VLM annotates images instead of the Oracle, SENSEI shows similar behavioral trends, but overall less object interactions, such that neither of the GPT-4 annotations on the Plan2Explore data completely match the performance of the oracle annotator.

Finally, when we compare the performance for SENSEI using GPT-4 annotations with two-angle camera images vs. only the right camera angle image, we see that the two-angle version performs better in terms of drawer interactions. This is expected since the drawer is more clearly visible in the left camera view. However, as the ball and blocks are mainly initialized on the right side of the table, the pure right camera angle SENSEI generates more interactions with these objects during exploration. Another factor here is that for the right camera angle we retain all 200K pairs for VLM-MOTIF distillation, whereas we only keep ca. 70% of the pairs in the case of SENSEI using both cameras for annotation.

## D.3    ROBODESK: REWARDS

In addition to interaction metrics, we count the number of times task rewards are collected during exploration. We observe that for the majority of tasks SENSEI solves more tasks in the environment during play compared to Plan2Explore. Note that for the `open_slide` task you need to open the slide fully in one direction, which is achieved in abundance in Plan2Explore runs by simply stretching the arm. The full interaction metrics of exploring how the slide moves left-right is not necessarily reflected in the task rewards, as can be seen in comparison to Fig. 5. Similar arguments also apply for opening the drawer fully vs. opening and closing the drawer more dynamically. Additionally as the bin is not really visible in our camera angle, solving `in_bin` tasks are more due to the objects that go off the table landing by chance in the bin for all methods, such that higher statistics for `off_table` rewards also lead to higher `in_bin` rewards.

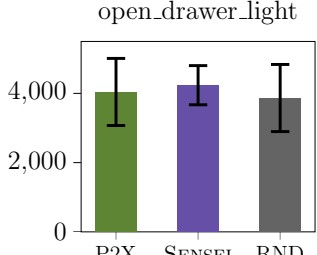

Figure 12: Collected rewards for `open_drawer_light` during exploration with SENSEI, P2X, and RND (3 seeds).

## D.4    ROBODESK: VLM-MOTIF WITH GENERAL PROMPT

In this section, we investigate the distilled reward function when using a general prompting strategy (SENSEI GENERAL, see Suppl. C.3.2). As shown in Fig. 14, the semantic reward $r_t^{\text{sem}}$ for the general prompt seems to show a high positive correlation or qualitatively matches with the VLM-MOTIF distilled using the specialized prompt in Robodesk (see Suppl. C.3.1). Thus, we manage to distill a reward function that peaks at interesting moments of exploration without injecting any environment specific knowledge into the prompt.

## D.5    ROBODESK: BASELINES

We present two new baselines in Robodesk: RND trained with PPO and pure VLM-MOTIF, and analyze the interaction metrics in Fig. 15. On average, SENSEI interacts more with most available objects than the baselines. RND mostly moves the arm around in the center of the screen, occasionally hitting objects or mostly buttons. It is important to note that the robot arm in Robodesk is mostly initialized close to the buttons. Pure VLM-MOTIF is an ablation of SENSEI without any information gain objective. Here, we see the importance of the information gain reward to ensure diverse exploration. Unlike SENSEI, we see that VLM-MOTIF interacts with specific entities: mostly the

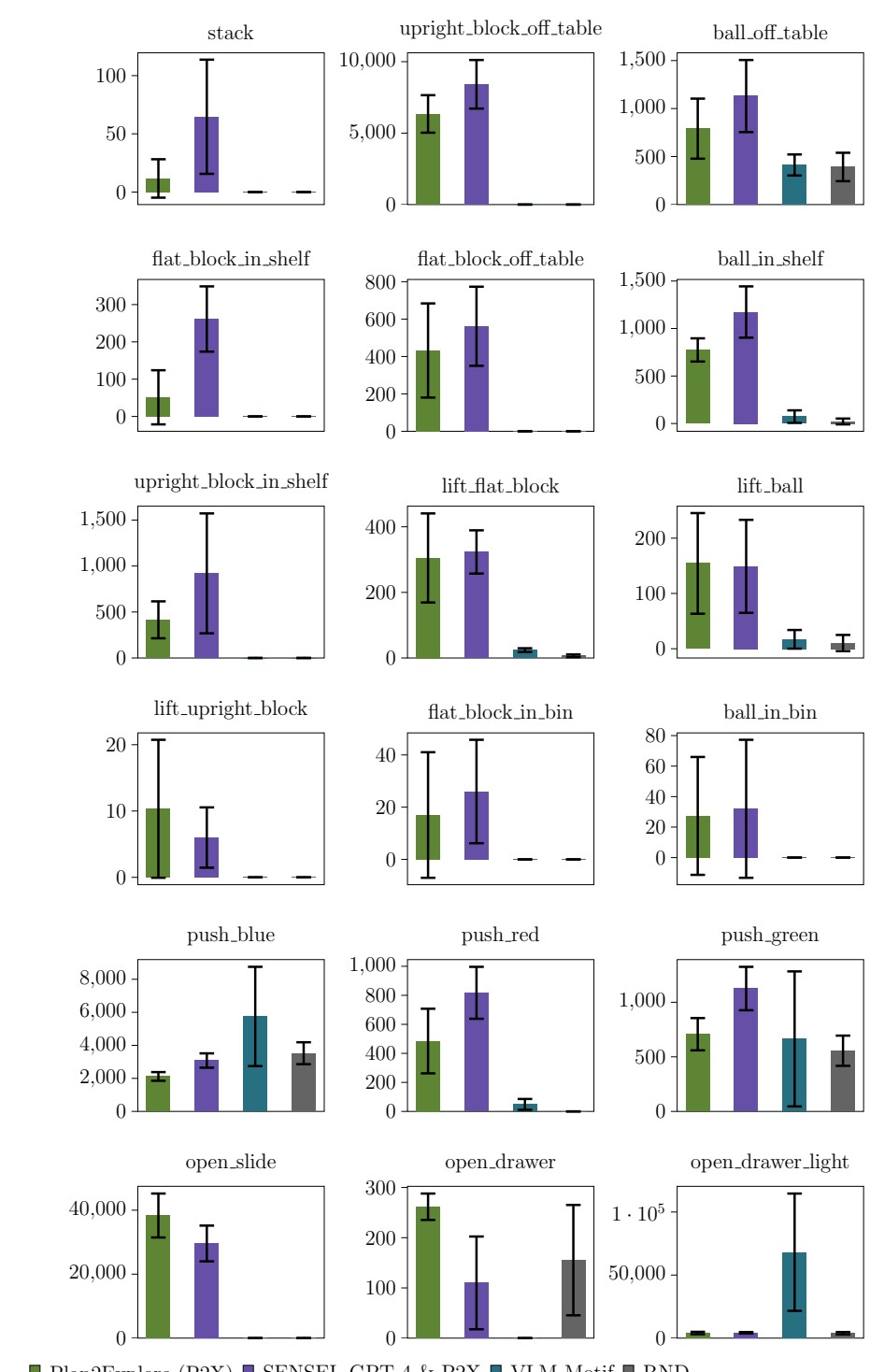

Figure 13: **Robodesk environment rewards**: We plot the mean number of sparse rewards (successful task completions) discovered during 1M steps of task-free exploration for all tasks for Plan2Explore, SENSEI, pure VLM-MOTIF, and the RND baseline.

buttons, the drawer and the flat block. The lack of interaction with the cabinet, the upright block and the ball are expected as these entities are spatially further away from the robot initialization pose. Once high semantic rewards are found in the vicinity by interacting with the drawer and buttons,

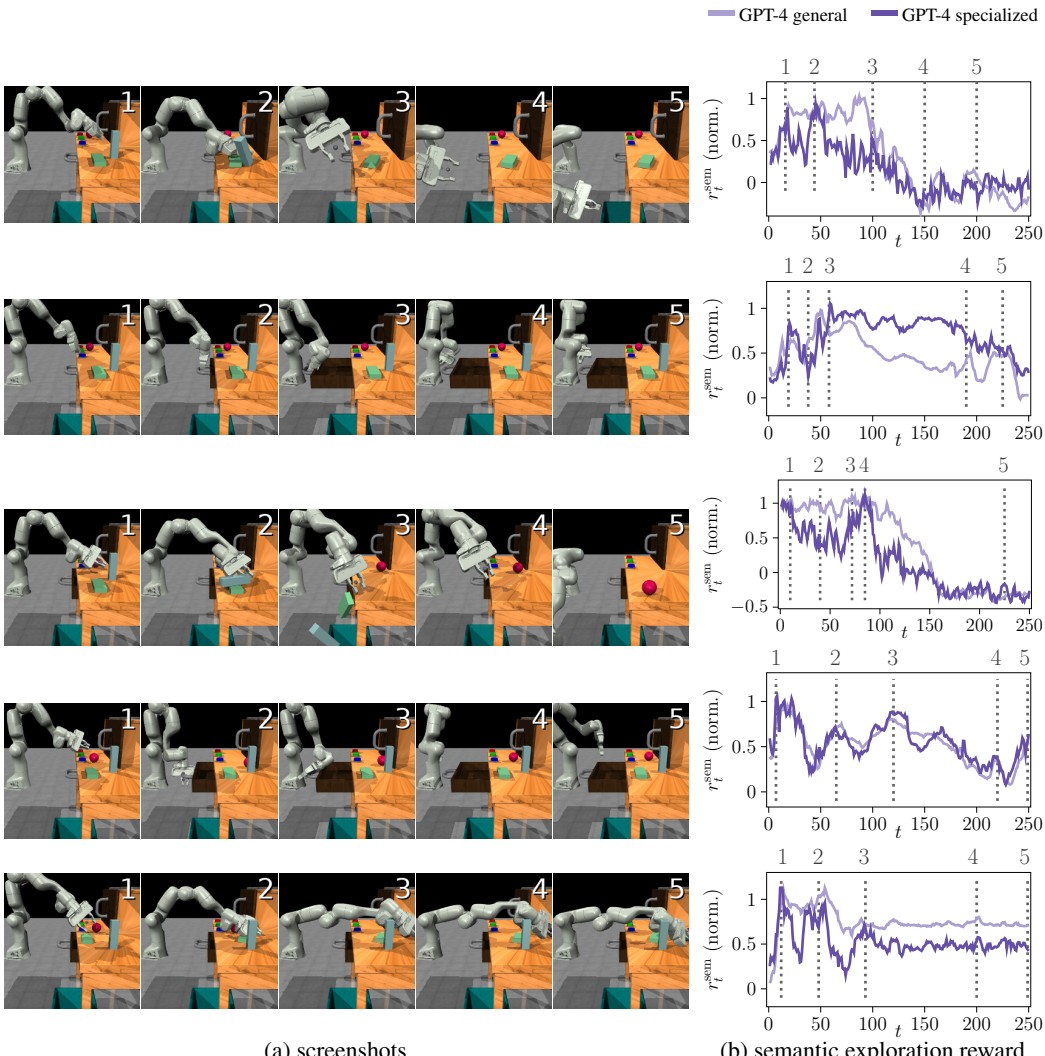

(a) screenshots          (b) semantic exploration reward

Figure 14: **Semantic exploration rewards for example trajectories with VLM-MOTIF using general vs. specialized prompts**: For five example Robodesk episodes, we showcase VLM-MOTIF semantic rewards distilled from GPT-4 annotations using a prompt specialized to the environment vs. a general prompt using multi-turn annotations using the same dataset (data from Plan2Explore runs). The reward trajectories for both the general and specialized prompts peak at the "interesting" moments of exploration, such as opening a drawer or pushing the blocks. With zero external knowledge injection, the general prompt version of VLM-MOTIF is highly correlated with its specialized prompt counterpart.

there is no incentive for pure VLM-MOTIF to explore further. On the other hand SENSEI aims to discover interesting and yet novel behaviors, ensuring better coverage across the different useful behaviors in the environment.

## D.6 ROBODESK: SENSEI WITHOUT DYNAMIC SCALING AND ANALYZING HYPERPARAMETER SENSITIVITY

In this section, we ablate the dynamic scaling of the semantic reward $r_t^{\text{sem}}$ and the information gain reward $r_t^{\text{dis}}$ terms in SENSEI. In SENSEI, we adjust the weight of these two terms based on whether $r_t^{\text{sem}}$ has reached the high percentile region of interestingness ($r_t^{\text{sem}} \geq Q_k$), as per equation Eq. 8. In this ablation, we instead use a linear combination with fixed weights $\alpha$ and $\beta$, such that the

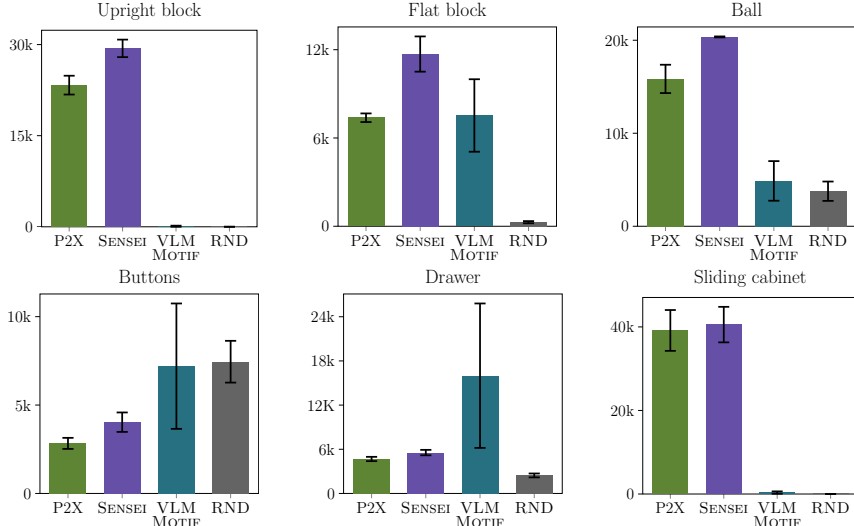

Figure 15: **Interactions in Robodesk**: We plot the mean over the number of interactions with any object during 1M steps of exploration for SENSEI, Plan2Explore (P2X), pure VLM-MOTIF and Random Network Distillation (RND) trained with a PPO policy, as a model-free exploration baseline. Error bars show the standard deviation (3 seeds).

exploration reward is given by:

$$r_t^{\text{expl}} = \alpha r_t^{\text{sem}} + \beta r_t^{\text{dis}}. \tag{9}$$

We present the results in Fig. 16 for 6 different sets of fixed weights. First of all, we observe that none of the fixed scale settings outperform SENSEI nor do they consistently perform as well as SENSEI. Second of all, we see that the exploration behavior is very sensitive to the choice of the weights $\alpha$ and $\beta$. For larger $\alpha$ values, the behavior collapses to mostly interacting with the drawer, buttons and the flat block, with larger fluctuations. This mode is very similar to the case of pure VLM-MOTIF presented in Fig. 15.

Next, we test the hyperparameter sensitivity of SENSEI with dynamic scaling of the reward weights. We see in Fig. 17, that across all 4 hyperparameter configurations, SENSEI is better or at least on par with Plan2Explore, and we don't observe any behavior collapse as in the fixed scale setting. We argue that although the dynamic scaling introduces additional hyperparameters, the overall behavior is much more robust and less dependent on hyperparameter tuning.

Table 1: Hyperparameter configurations for SENSEI presented in the main experiments and the 3 other configurations that are shown in Fig. 17.

|  | SENSEI | SENSEI HP1 | SENSEI HP2 | SENSEI HP3 |
|---|---|---|---|---|
| Quantile | 0.75 | 0.80 | 0.85 | 0.75 |
| $\alpha^{\text{explore}}$ | 0.1 | 0.01 | 0.1 | 0.05 |
| $\beta^{\text{explore}}$ | 1 | 1 | 1 | 1 |
| $\alpha^{\text{go}}$ | 1 | 1 | 1 | 1 |
| $\beta^{\text{go}}$ | 0 | 0 | 0 | 0 |

## D.7 COMPUTATION

SENSEI has 3 phases: (1) annotation of data pairs (offline), (2) reward model, i.e. VLM-MOTIF, training (offline), (3) online RL training with environment interactions (DreamerV3). All experiments were performed on an internal compute cluster.

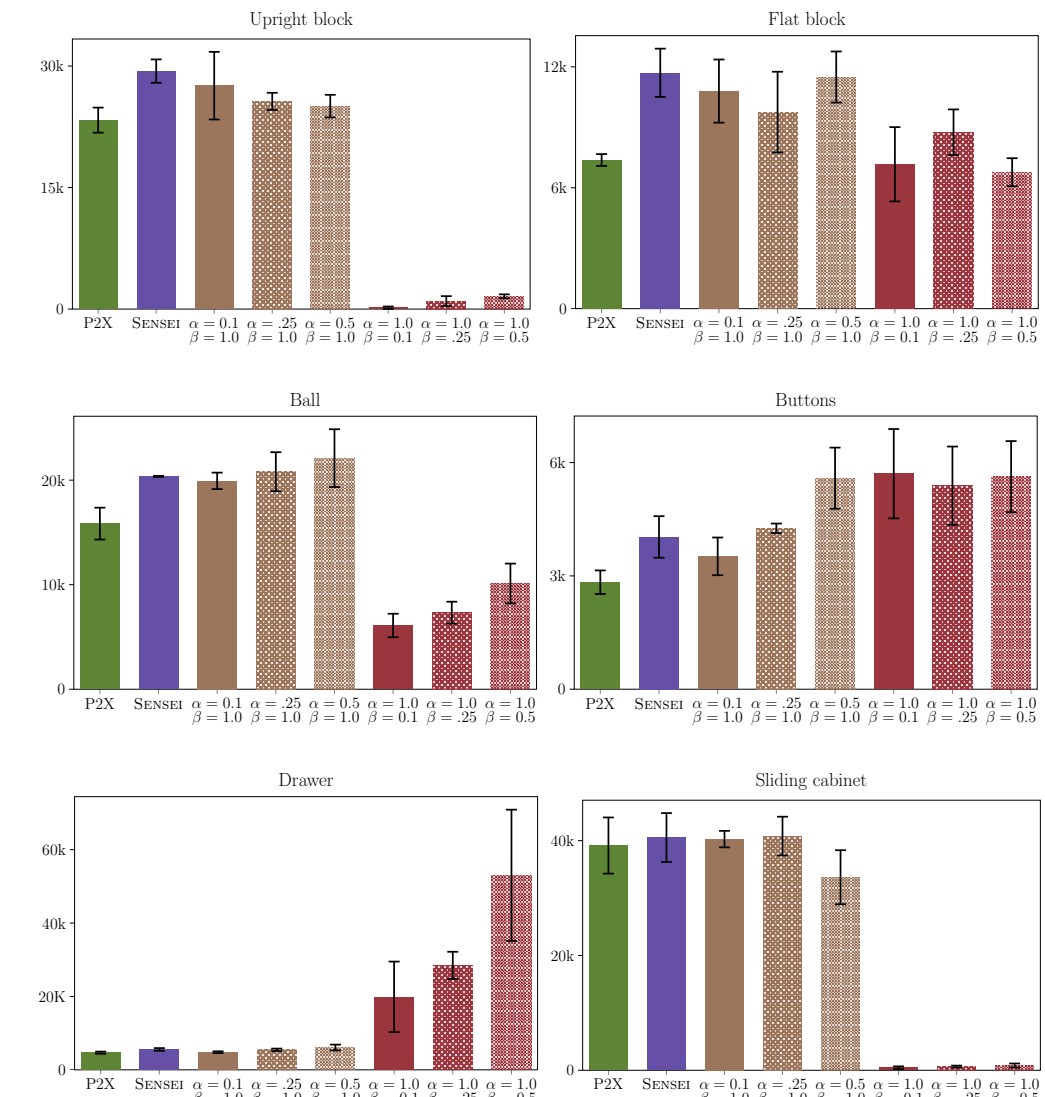

Figure 16: **Comparing Interactions in Robodesk between SENSEI and Fixed Scaling of Rewards**: We plot the mean over the number of interactions with any object during 1M steps of exploration for SENSEI and Plan2Explore (P2X) and an ablation of SENSEI, where we do not dynamically adjust the weight of the reward terms based on the current semantic reward. For this ablation, reward is computed as $r_t^{\text{expl}} = \alpha r_t^{\text{sem}} + \beta r_t^{\text{dis}}$ with fixed weights $\alpha$ and $\beta$. Error bars show the standard deviation (3 seeds).

**Dataset Annotation**    The annotation of data pairs is done using the OpenAI API, such that a single CPU is sufficient. For instance for Robodesk with a dataset size of 200K pairs, we parallelized this over 200 CPUs, where we annotated 1K pairs each, which took approximately 40 minutes. Note that annotations are fully offline and do not affect the runtime of SENSEI itself. Each annotation using the single-turn strategy cost $0.002 with `gpt-4o-2024-05-13` and $0.004 with `gpt-4-turbo-2024-04-09`. The multi-turn prompting for the zero-knowledge Robodesk annotations also cost $0.004 per pair with `gpt-4o-2024-05-13`.

**Reward Model Training**    After annotating the dataset, we train the VLM-MOTIF network using a single GPU for 50 epochs. Using e.g. Tesla V100-SXM2-32GB, this took 20min. We ran a grid search over different hyperparameters for VLM-MOTIF training (batch size, learning rate, weight

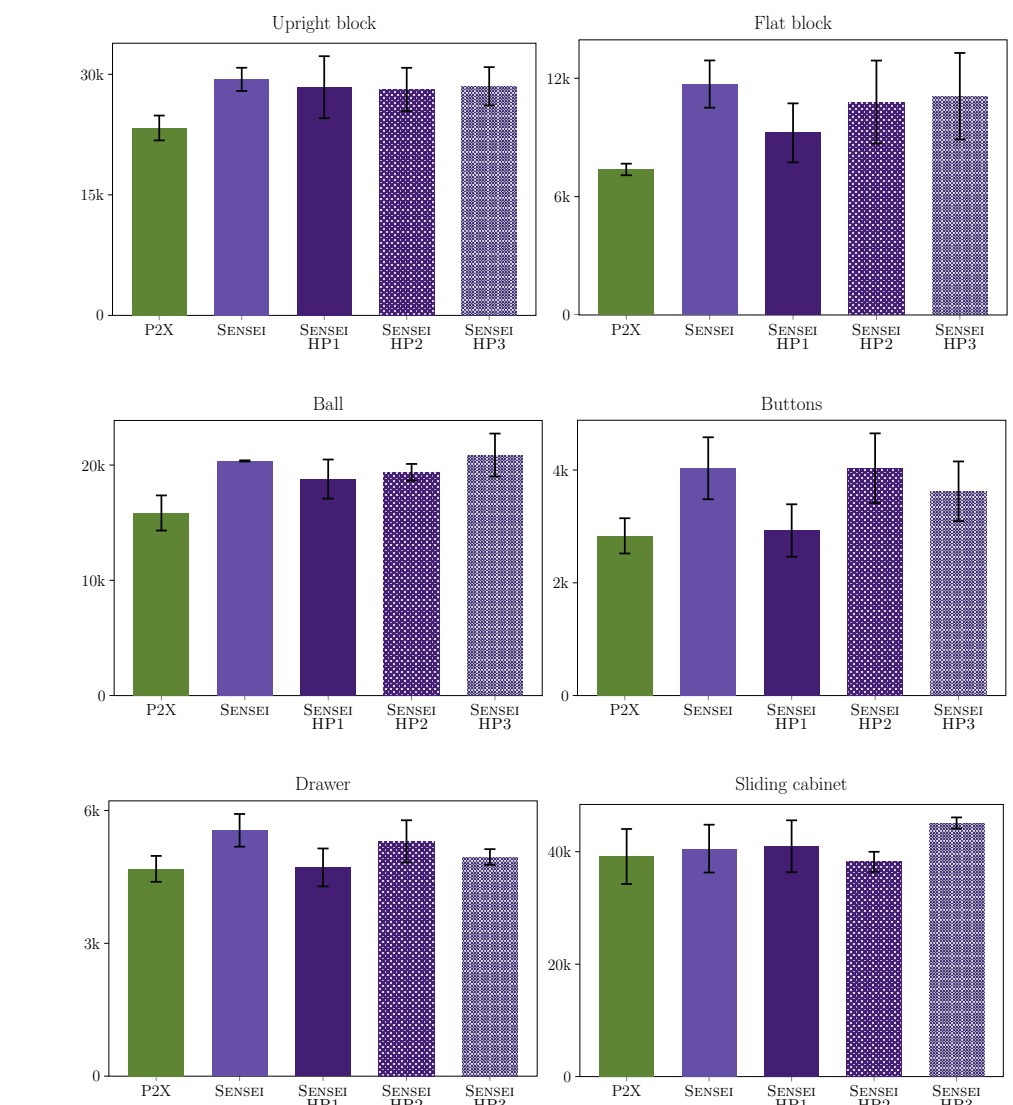

Figure 17: **Comparing Interactions in Robodesk for SENSEI with different hyperparameters**: We plot the mean over the number of interactions with any object during 1M steps of exploration for SENSEI (winner hyperparameter configuration) and Plan2Explore (P2X) and SENSEI with different hyperparameters as specified in Table 1. Error bars show the standard deviation (3 seeds).

decay, network size), testing for a total of 18 different combinations, and we chose the reward model with the best validation loss to use in SENSEI runs.

**Online Model-based RL Training**     SENSEI is built on top of DreamerV3, just like our main baseline Plan2Explore. On a NVIDIA A100-SXM4-80GB, SENSEI runs at ca. 7.5Hz, Plan2Explore runs at ca. 10Hz and pure VLM-MOTIF runs at ca. 8.7Hz.

