# OpenReview forum: "SENSEI: Semantic Exploration Guided by Foundation Models to Learn Versatile World Models"
_ICLR.cc/2025/Conference — Submitted to ICLR 2025_

### Official Review · Reviewer_dZbp · 2024-10-17

**Soundness:** 4
**Presentation:** 4
**Contribution:** 2
**Rating:** 6
**Confidence:** 4

**Summary:**

The authors introduce SEmaNtically Sensible Exploration (SENSEI), a technique inspired by the previous work MOTIF, which uses Large-Language Models (LLMs) to express preferences between different observations in order to generate a reward model for a (model-free) reinforcement learning agent. The authors propose adapting this technique for Vision-Language Models (VLMs) and integrating it with world model learning. After an initial data collection phase, an "interestingness" reward model is distilled from the VLM’s preferences over image observations. This reward model is combined with the Recurrent State-Space Model (RSSM) of DreamerV3 and the ensemble technique used in Plan2Explore to develop an exploration policy that is influenced by the learned semantic reward. The results demonstrate that this combination of VLM-MOTIF and Plan2Explore leads to more effective exploration than relying on either technique alone. Additionally, downstream task experiments show that SENSEI outperforms baseline methods in terms of task performance.

**Strengths:**

- The extension of MOTIF to VLMs feels natural and improves the applicability of such techniques beyond environments with text representations.
- The paper does not solely rely on the semantic reward model but recognizes that epistemic uncertainty is important to consider. The authors analyse this and propose a well-motivated adaptive solution that combines the two sources of information to achieve more effective exploration.
- The paper provides a solid empirical validation; the experiments are overall well-explained and detailed and show that SENSEI outperforms baseline approaches in terms of exploration quality and downstream task performance

**Weaknesses:**

- The proposed method seems to rely on domain-specific prompt engineering. For instance, as shown in Appendix C3, in the Robodesk environment, the VLM is provided with detailed, manually crafted prompts specifying what constitutes "interesting" behavior. This significantly limits the generality and practicality of the proposed method.
-  The method heavily relies on the dataset collected through a self-supervised exploration technique, which is then used for annotation and reward model training. A key concern here is that the distilled reward model is inherently biased towards the specific data collected through this relatively simple exploration technique. This may not pose issues in small, simple environments but is likely to be fragile in more complex, large-scale environments. The reward model is static, meaning it cannot adapt to observations outside the distribution of the pre-collected dataset, limiting its ability to generalize to novel situations.
- In the Appendix, it is mentioned that the authors used 139409 pairs of observations for Robodesk. If I understand correctly, this means the VLM was prompted 139409 times to obtain the annotations for a single environment. If so, the practicality and scalability of the approach becomes slightly concerning. The cost of the experiments is not mentioned in the paper, but it is likely that querying VLMs this often could become computationally expensive, particularly in larger-scale environments.

**Questions:**

**Suggestions**
- Regarding the reliance on detailed prompt engineering, one potential resolution could be to lean more heavily on the prior knowledge and inference capabilities of the VLM. Instead of providing explicit descriptions of the environment, the VLM could be tasked with inferring the context from the visual observations. This would allow the model to deduce what constitutes interesting behavior without needing environment-specific prompts. While this might sacrifice some performance, it could provide a more general solution applicable across a variety of visual environments without manual prompt engineering for each domain.
- A suggestion regarding the concern about over-reliance on the initial pre-collected dataset in larger, more complex environments would be to consider introducing additional phases of data collection and retraining. After the initial exploration and training with the distilled reward model, the agent (using DreamerV3 + SENSEI) could collect new data during subsequent exploration, which could then be annotated and used to retrain a new reward model. Although this approach would still be biased by the initial dataset, I believe that this direction of research could progressively reduce the bias and help the model better adapt to novel observations.

**Minor comments / Questions**
- Line 146, small error 'In reward training phase'
- I would suggest omitting the parts from your section titles in Section 2, such as "UNLEASH YOUR SENSEI", as it does not fit the professional tone of the rest of the paper.
- At a high level, in your proposed exploration scheme, the agent is incentivized to first find interesting states and then switch to an uncertainty-maximizing approach. However, what is not completely clear to me is how or when it then switches back to $\beta_{Go}$ to continue finding new interesting states?
- In Figure 5, why is there no comparison with VLM-MOTIF, whereas in Figure 4, there is?
- In Figure 4, the SENSEI agent performs poorly in the "at chest with key" scenario compared to VLM-MOTIF. You mention that this is because "being at the chest with a key is an 'interesting' state, so there is no real incentive for the agent to explore what would happen if the chest was opened." However, intuitively, one might expect that an open chest would be considered even more interesting. Is there any further intuition as to why the VLM preferences do not guide the agent in exploring this further? Have you seen in annotations that if presented as a pair, the key+chest was rated more interesting than an open chest?

---

> ### Author Response · Authors · 2024-11-24
> **Response to Reviewer dZbp [1/2]**
>
> Thank you for your thorough read, your great comments and your helpful suggestions.
>
> ## Domain specific prompting
>
> Thank you for raising this point and the fantastic suggestion that the “VLM could be tasked with inferring the context from the visual observations”. We have now added a new experiment where we perform annotations in Robodesk without injecting any environment specific knowledge. We instead follow a multi-turn prompting strategy, similar to your suggestion, where we first prompt the VLM with an image of the environment, asking the VLM to describe it in detail. We then ask for it to rank two images based on its own description of the environment (prompt details in Supp. C.3.2). Finally, we empirically show that the semantic rewards of the general prompt version of VLM-MOTIF with zero external knowledge injection behaves qualitatively very similar to its environment specialized prompt counterpart and matches the peaks in “interesting” moments of interactions  (Supp. D.4, Fig. 14).
>
> ## Reliance on initial dataset
>
> It is true that SENSEI reinforces trends in the initial dataset used for VLM annotation, which we analyze for Robodesk in Suppl. D.2. We agree with your analysis of multiple rounds of SENSEI annotations as a remedy, as we had also discussed in our future work section. We believe this is an exciting research direction, and we are happy to hear that you share our enthusiasm.
>
>
> ## Computational cost of experiments
> It is correct that, for the case you described, the VLM is prompted roughly 140k times. This is indeed computationally costly. However, these computations are completely offline, and do not need to happen at runtime of SENSEI such that they can be parallelized. Furthermore, because the VLM does not need to be prompted while training the world model, SENSEI is relatively efficient during runtime and runs on just one GPU. We add a new section detailing the computational cost of our experiments in Suppl. D.7.
>
> ## Comments and suggestions
>
> Thank you for the careful reading and catching errors.
>
>  > I would suggest omitting the parts from your section titles in Section 2, such as "UNLEASH YOUR SENSEI"
>
> We changed this title as it is indeed a bit unprofessional. But we kept some other section titles that we felt could help structure the flow of this section and are not too unprofessional.
>
> > However, what is not completely clear to me is how or when it then switches back to continue finding new interesting states?
>
> The switching mechanism is implemented by keeping statistics of the predicted interestingness of states. A quantile of these statistics, $Q_k$ for the $k$th quantile, is used to switch between a mode favoring uncertainty-based exploration ($\beta^\mathrm{explore}$) or semantic exploration ($\beta^\mathrm{go}$).  As long as the agent is in an interesting state, with $\hat{r}^{sem}_t \geq Q_k$ , it keeps exploring with high uncertainty maximization ($\beta^\mathrm{explore}$). Only when the agent reaches a state that is not interesting anymore, indicated by $\hat{r}^{sem}_t < Q_k$, it more strongly favors exploration based on semantic rewards ($\beta^\mathrm{go}$).
>
> Let’s consider the Robodesk setting as an example. A typical observation might have the robot just move the gripper across or over the table. However, when the gripper is near the ball on the table, this could be a more interesting state than what it typically encounters ($\hat{r}^{sem}_t \geq Q_k$). The agent would try actions that cause high uncertainty. When these actions lead to pushing the ball off the table, the agent ends up in an uninteresting state again with no object nearby to interact with ($\hat{r}^{sem}_t < Q_k$). Thus, it strives to mainly maximize semantic rewards again, for example by moving the gripper to the next available object.
>
> We thank the reviewer for allowing us to further clarify this and we have now expanded our description of this mechanism in the main paper.
>
> > In Figure 5, why is there no comparison with VLM-MOTIF, whereas in Figure 4, there is?
>
> We add this baseline now in Suppl. D5. We show that SENSEI outperforms VLM-Motif in most interactions metrics (Fig. 15) and rewards (Fig. 13), showcasing the importance of the information gain objective of SENSEI in the Robodesk environment as well.

---

> > ### Author Response · Authors · 2024-11-24
> > **Response to Reviewer dZbp [2/2]**
> >
> > > You mention that this is because "being at the chest with a key is an 'interesting' state, so there is no real incentive for the agent to explore what would happen if the chest was opened." However, intuitively, one might expect that an open chest would be considered even more interesting.
> >
> > Importantly in MiniHack there is no sprite for an open chest. Instead after opening the chest the episode ends. This is an important part that was missing from our original explanation, which we now added.
> >
> > To give a more detailed explanation for this example: When the agent has reached the chest with a key, if it is optimizing purely for $r^{sem}_t$, the agent just staying at the chest without doing anything will give high semantic rewards. There is no incentive to try novel actions, such as executing the OPEN action. Even if the agent by chance opens the chest once, it only learns that this will decrease interestingness because the episode terminates. Thus, in the future this action is less likely to be repeated. On the other hand, when SENSEI has reached the chest it tries many new actions, including trying to open the chest from all possible directions while it still is not fully certain about this outcome, as it also tries to maximize an information gain reward.
> >
> > We hope we could clarify all open questions and would like to thank the reviewer again for their careful read and comments.

---

> ### Comment · Area_Chair_6YtS · 2024-11-25
> **Please read rebuttal**
>
> Dear Reviewer dZbp, Could you please read the authors' rebuttal and give them feedback at your earliest convenience? Thanks. AC

---

> > ### Comment · Reviewer_dZbp · 2024-11-26
> >
> > I appreciate the author's clarifications and additional experiments that were conducted. I would not be opposed to this paper being accepted as I think it can be considered sufficiently interesting for ICLR. However, I will not raise my score as it remains relatively incremental in terms of novelty and performance is heavily reliant on the initial dataset that is collected.

---

### Official Review · Reviewer_5Hj6 · 2024-10-21

**Soundness:** 3
**Presentation:** 3
**Contribution:** 3
**Rating:** 8
**Confidence:** 4

**Summary:**

The authors propose SENSEI, a framework to equip model-based RL agents with intrinsic motivation for semantically meaningful behavior.
They distill an intrinsic reward signal of interestingness from VLM annotations, following MOTIF. The agent learns to predict and maximize these intrinsic rewards using a world model, using RL algorithms similar to Dreamerv3. They conduct experiments on both robotic and
video game-like simulations and achieve good results.

**Strengths:**

1. Good motivation and experiments. The authors explain their methods very clearly, and their experiments are diverse and convincing. Incorporating VLM into reinforcement learning is a hot topic, but this kind of integration sounds novel.

2. Various related work. The coverage of the huge amount of related work is about as thorough as possible.

3. Sufficient ablation study. The authors conduct several ablations to demonstrate the effectiveness of each component.

**Weaknesses:**

1. It seems that most components come from existing work.

2. MOTIF uses much weaker VLMs, so it's possible that the performance gain mostly come from the improvement of VLMs. The authors could perform experiments to demonstrate the extent to which performance declines when using a less powerful model.

**Questions:**

1. The authors may briefly explain how long it will take to train SENSEI (dataset annotation, reward training and RL training) and the baselines using the same computation resources.

2. The authors emphasize a realistic setting. It would be much harder to create a 200K dataset of preferences if we want to deal with real world embodied agents. Have the authors tried to use a smaller dataset?

---

> ### Author Response · Authors · 2024-11-24
> **Response to Reviewer 5Hj6**
>
> We thank the reviewer for the excellent feedback. We gladly answer the open questions:
>
> ## Do SENSEI’s improvements mainly come from VLM size increase?
>
> It is true that Motif relies on a smaller LLM than GPT-4 used in our work. However, the improvements of SENSEI cannot only be attributed to scaling foundation models, but the way how semantic rewards are adaptively combined with uncertainty-based exploration.
> For example, our VLM-Motif baseline also uses GPT-4 annotations. Nonetheless it rarely discovers environment rewards during exploration (see Fig. 4).
> SENSEI, on the other hand, uses the semantic knowledge distilled from VLM annotations as a starting point for exploration and then branches out to discover new behavior.
> We also want to highlight that Motif in the original work is only text-based. Even though we have an increase in model size in our case, the difficulty of annotations is also significantly increased in our problem setting since our annotations are image-based, requiring spatial grounding capabilities from the VLM.
>
> ## Training resources
> Thank you for the great suggestion. We add a new section in which we detail all computational resources needed to train SENSEI in Supp. D.7. for all stages of SENSEI.
>
> ## Smaller training data
>
> This is a good point that we would like to share our insights on. Especially in environments with rich and high-dimensional observations with a potentially long-tailed distribution, such as in Robodesk, we believe more data is useful to make sure we don’t run into many out-of-distribution (OOD) states while running SENSEI. We see signs of this already in some of our experiments. For the right camera only ablation in Robodesk (Supp D.2), we keep the entire dataset with 200K samples. Since the annotations use a single camera angle, e.g. the drawer can be occluded, this can lead to false annotations from the VLM. That’s why in our main experiments, we annotate the same dataset of 200K pairs also with the left camera angle and only keep the pairs where both annotations agree. Doing so we only keep 69% of the original data, which is then used for Motif training. As a result, our annotations with dual-cameras are much less noisy, leading to improvements in drawer interactions compared to the right-camera only variant. However, we don’t yet match the performance of an oracle annotator, or even the right camera angle for some objects. We hypothesize this is also in-part due to the data we filter out in the process, increasing chances of OOD states and thus noise in semantic rewards during exploration with SENSEI. We expect however that the OOD issue to not be as significant in Minihack environments, where the observations are not as rich and diverse as Robodesk.
>
> In order to decrease dataset sizes for Motif training without increasing OOD risk in Robodesk-like environments, we could try to ensure better coverage in the dataset used for reward model training. We believe this ties in nicely with our motivated future work, where we would use the buffer generated from a SENSEI run, that has richer interactions, to bootstrap a new Motif network.
>
> We hope we could clarify all open questions and would like to thank the reviewer again for their feedback.

---

> > ### Comment · Reviewer_5Hj6 · 2024-11-25
> >
> > I believe the author has mostly addressed my question. At the same time, I have also taken note of the issues raised by other reviewers, such as the accessibility of interestingness and domain-specific prompts. I find the author's responses to these concerns to be fairly reasonable as well. I will maintain my score of 8.

---

> ### Comment · Area_Chair_6YtS · 2024-11-25
> **Please read rebuttal**
>
> Dear Reviewer 5Hj6, Could you please read the authors' rebuttal and give them feedback at your earliest convenience? Thanks. AC

---

### Official Review · Reviewer_71Ub · 2024-10-30

**Soundness:** 2
**Presentation:** 3
**Contribution:** 2
**Rating:** 6
**Confidence:** 4

**Summary:**

This paper proposes a way to combine VLM feedback and exploration bonuses. The main idea of this paper is to combine Plan2Explore, a latent ensemble disagreement-based exploration method, with a MOTIF-based intrinsic reward that tells the agent how interesting a state is. Specifically, SENSEI first trains a semantic reward model based on a dataset using MOTIF (w/ preference-based reward learning), and then runs Plan2Explore augmented with the learned semantic reward function. The authors also propose an additional mechanism to adaptively control the coefficients for the semantic and ensemble-based intrinsic rewards. They show that their method (SENSEI) outperforms Plan2Explore and VLM-MOTIF on MiniHack and Robodesk tasks.

**Strengths:**

* The proposed method is straightforward and reasonable.
* The authors empirically show that SENSEI improves exploration on both MiniHack and Robodesk, and the paper has some analysis results as well.
* The paper is generally well-written. In particular, I enjoyed reading Sections 1 and 2.

**Weaknesses:**

* The method is largely a straightforward combination of two existing techniques: Plan2Explore and MOTIF. While I think this *alone* doesn't necessarily constitute a ground for rejection, I do think the results are not terribly surprising nor extremely insightful in the current form. It may have been more informative to practitioners if the authors had focused much more on ablation studies or analyses (beyond those in Appendix D). For example: Is MOTIF the only way to define a semantic reward model, and if so, how/why is it better than other alternatives? Is having two phases necessary (can we not have a separate pre-training stage)? Is the adaptive coefficient adjusting strategy (Eq. (7)) necessary? How sensitive is the performance to this strategy?
* The experimental results are somewhat limited. The authors only compare SENSEI to its ablations (P2X, VLM-MOTIF), and it is unclear how SENSEI performs compared to other types of exploration methods like LEXA/PEG, or VLM-based exploration methods like ELLM/OMNI/LAMP. I don't expect comparisons with all of these baselines, but I think it is important to have at least some comparisons with different types of previous methods in each category.
* The hyperparameter table in Appendix B implies that SENSEI is potentially sensitive to various hyperparameters. For example, it seems SENSEI requires careful, individual tuning of the quantile hyperparameter for each individual task in Robodesk (it uses 0.75, 0.75, 0.80, and 0.85 for each task). How were these values chosen, and did the authors perform the same level of hyperparameter tuning for the baselines as well? Did the authors decouple the runs for hyperparameter tuning and for the results (the results will otherwise be biased)? How sensitive is SENSEI to these individually tuned hyperparameters (quantile, batch size, learning rate, weight decay, etc.)?
* The authors use only 3 seeds for the experiments, which further makes it difficult to evaluate the empirical contributions of the method.

**Questions:**

Please answer the questions in the weaknesses section above. The paper is mostly clear, and I don't have any other specific clarification questions.

---

> ### Author Response · Authors · 2024-11-24
> **Response to Reviewer 71Ub**
>
> We thank the reviewer for their valuable feedback and great suggestions.
>
> ## Coefficient adaptations & Hyperparameter sensitivity
> Thank you for this excellent suggestion to ablate the coefficient adaptation strategy. In our new supplementary experiment in Robodesk, we test a version of our method with fixed reward weights (Supp D.6)
>
> We present the results in Fig. 16 for six different sets of fixed weights. First of all, we observe that none of the fixed scale settings outperform SENSEI, nor do they consistently perform as well as SENSEI. Secondly, we see that the exploration behavior is very sensitive to the choice of these fixed weights. E.g., for larger weights on the semantic reward, the behavior collapses to mostly interacting with only specific objects.
>
> In conjunction to this, we also tested the hyperparameter sensitivity of SENSEI when using dynamic scaling of the reward weights. We showcase in Figure 17 that across different hyperparameter configurations, SENSEI’s behavior is much more robust and is better or at least on par with Plan2Explore in all cases. We don't observe any behavior collapse, in contrast to the fixed scale setting. We would therefore argue that the overall behavior of the dynamic scaling  is much more robust and less dependent on hyperparameter tuning compared to fixed reward coefficients.
>
> ## Exploration baselines
> Thank you for your baseline suggestions. We have added a new exploration baseline to further demonstrate the advantage of using SENSEI. In Robodesk we now also compare SENSEI to Random Network Distillation (RND) [ref 1], a popular exploration strategy that uses the prediction errors of random embeddings of input images as an intrinsic reward to guide a policy towards unseen regions. Except for button presses, SENSEI interacts with all objects more frequently than RND and, as a result, discovers more rewards during exploration.
>
>
> Regarding the other suggested baselines – Unfortunately we do not think these are particularly suited or applicable for the setup we consider.
> As discussed in Section 3, ELLM and OMNI require text-based environments or a mapping from environment states to text, where OMNI also assumes access to reward functions for all potential tasks in the environment during training. On the other hand, SENSEI is designed for environments with pixel-based inputs without assuming access to environment task rewards during exploration.
>
> LEXA is a goal-conditioned extension of Plan2Explore: after an exploration phase with Plan2Explore, in a second stage, LEXA randomly samples goals from its replay buffer to train a goal-conditioned policy. While applicable in our environments, the exploration phase in LEXA is Plan2Explore. We see LEXA’s addition of goal-conditioned RL to the Dreamer framework as an orthogonal research direction to our work that could also be applied to SENSEI.  For example, Plan2Explore exploration phases in LEXA could be replaced by exploration with SENSEI, and the uniform sampling from the replay buffer can be replaced by sampling from the top-k samples, as ranked by our VLM-Motif reward.
>
> PEG extends LEXA by a more sophisticated exploration by searching for exploration goals likely to lead to high epistemic uncertainty. Unfortunately, PEG performs its goal search in observation space. Thus, it requires lower dimensional observations (positions) and is not applicable to our pixel-based environments.
>
> LAMP proposes a pre-training strategy for a language-conditioned policy, given a diverse set of tasks generated by hand or by an LLM. This means that e.g. for Robodesk, LAMP in the pre-training phase already tries to optimize for tasks such as “open the drawer”, “lift the block”. Compared to our work, the focus is not on discovering useful behaviors to engage with in a given environment, but on trying to learn a given set of useful behaviors better. Given this discrepancy in the method’s objectives and experimental setup, it is unclear to us how to ensure a fair comparison between them.
>
>
> ## Hyperparameter sensitivity
> As mentioned above, we overall found the behavior of SENSEI to be robust to hyperparameters, and did not observe behavior collapse in any of the settings we tested for. As a general rule, we did perform a hyperparameter grid search for all of our experiments (SENSEI and all baselines), and presented the best results we obtained in each case, maintaining a fair optimization budget over all tested methods.
>
> ## Seeds
> As per your suggestion, we now increased the number of random seeds to 5 in all Minihack experiments. For the final version of our paper, we plan to further increase the number of seeds and also add more seeds to our Robodesk experiments.
>
> ## References
> [ref 1] Burda, Yuri, et al. "Exploration by random network distillation.", ICLR 2019.

---

> ### Comment · Area_Chair_6YtS · 2024-11-25
> **Please read rebuttal**
>
> Dear Reviewer 71Ub, Could you please read the authors' rebuttal and give them feedback at your earliest convenience? Thanks. AC

---

> > ### Comment · Reviewer_71Ub · 2024-11-25
> >
> > Thanks for the response. I carefully went through the newly added sections and results, but I'm still not fully convinced of the empirical significance of this method. Despite the new results, many experiments still only use 3 seeds and have high variances, the error bars overlap in many cases, and alternative strategies are not sufficiently considered. I also feel this paper doesn't make substantially novel insights.
> >
> > That being said, this paper also has some strengths: the method is conceptually reasonable, the authors put a fair amount of effort into validating the method to some degree, and there are no obvious weaknesses. Compared to previously accepted papers at ICLR, I feel this work is right on the borderline. While I raised my score to 6 in acknowledgment of the new results, I would have given a 5.5 (borderline) if such an option existed, and wouldn't be opposed to rejection either.

---

> > > ### Author Response · Authors · 2024-11-28
> > > **Regarding experiment seeds and high variances**
> > >
> > > Thank you for raising your score.
> > >
> > > We understand your concern about empirical significance and high variances. To address this we have added the **interaction statistics of MiniHack for 10 random seeds** in Fig. 4. As you can see SENSEI still clearly outperforms the baselines in accumulated rewards and shows more favorable interaction statistics than the baselines (e.g., more often using the key to open the door).
> > >
> > > With more seeds we noticed that, especially in MiniHack-KeyroomS15, the variance of SENSEI tends to decrease, while the variance of Plan2Explore remains high. We believe that the trajectory of Plan2Explore can strongly vary depending on the randomly generated map and random starting positions. SENSEI is less susceptible to this environment randomness because it not only seeks out uncertain situations but also semantically meaningful interactions, like picking up a key and using the key.
> > >
> > > We are now running all experiments of the main paper with 10 random seeds for each configuration and baseline. Unfortunately not all runs have finished today. Thus, when all the experiments are finished, we will upload the updated plots (Fig. 5 & Fig. 6) to our supplementary website (see footnote of the first page of our paper). We will gladly inform you when the website is updated.
> > >
> > > We refer the reviewer to our new general response for new experiments and further updates in the paper.

---

> > > > ### Author Response · Authors · 2024-12-02
> > > > **Updated plots with more seeds**
> > > >
> > > > We wanted to notify that we have updated the plots on our webpage after running our experiments with 10 seeds. The trends hold as before and compared to our baselines, SENSEI interacts on average more with the relevant objects in Robodesk (Fig. 5) and is more sample efficient in learning to reliably solve the tasks in MiniHack (Fig. 6).

---

### Official Review · Reviewer_qRFT · 2024-10-31

**Soundness:** 2
**Presentation:** 3
**Contribution:** 2
**Rating:** 3
**Confidence:** 4

**Summary:**

This paper presents SENSEI, a model-based RL framework that guides agents' exploration by eliciting a human notion of "interestingness" from a vision-language model (VLM). The VLM annotates preferences between pairs of observations, gathered through self-supervised exploration. This annotated preference dataset is then used to train a reward model, SENSEI, which is subsequently distilled into the agent’s world model. The proposed approach demonstrates superior performance over baselines in experimental settings, including robotic and video game-like simulations.

While the paper introduces promising ideas, I believe it needs further development before it is ready for publication.

**Strengths:**

- Exploring how human priors can be integrated into RL agents using VLMs is a valuable research direction.
- The combination of intrinsic rewards derived from both VLMs and epistemic uncertainty effectively enhances performance.
- The graphical illustrations are clear and helpful, aiding readers in understanding the methodology and results.

**Weaknesses:**

- The primary concern is the incorporation of a human notion of "interestingness" in the decision-making process. I believe environments where humans can clearly express preferences between observations based on interestingness are limited. For instance, in settings like the game of Go or simple mazes with few interactive objects, it can be challenging to determine what constitutes an "interesting" observation. The authors attempt to address this in the prompt design described in Appendix C.3, where they define "interestingness" for specific environments. However, defining “interestingness” specifically for each environment may limit the method’s ability to incorporate true human priors. In my opinion, the actual prompt the authors use is simply a more specified version of that used in [1], which prompts for the option “most likely to make progress towards the goal” or “most likely to show improvement with respect to the goal.” As a result, the distinction between this work and previous methods may be less substantial. To enhance originality and better substantiate the claims, I recommend refining the prompt design to more clearly elicit human priors from VLMs and including a thorough ablation study on different prompt choices.
- Additionally, I found the formulation unnecessarily complex. The VLM preferences are distilled twice, first into the reward model (SENSEI) and then into the world model. A more straightforward approach might be to incorporate the reward model directly into the world model to simplify the structure and better convey the core idea.

[1] Martin Klissarov, Pierluca D’Oro, Shagun Sodhani, Roberta Raileanu, Pierre-Luc Bacon, Pascal Vincent, Amy Zhang, and Mikael Henaff. Motif: Intrinsic motivation from artificial intelligence feedback. The Twelfth International Conference on Learning Representations, 2024.

**Questions:**

- There is only one baseline in the main experiment. This paper could benefit from including more baselines to enable a more comprehensive comparative analysis.

---

> ### Author Response · Authors · 2024-11-24
> **Response to Reviewer qRFT [1/2]**
>
> Thank you for the feedback and raising some important questions.
>
> ## Criticism on human bias of interestingness for environments
>
> > I believe environments where humans can clearly express preferences between observations based on interestingness are limited
>
> We agree that for abstract environments, such as simple mazes or abstract games, it is unclear whether a pretrained VLM could extract a human bias of interestingness from screenshots alone. However, our long-term goal with SENSEI is to tackle realistic scenarios, such as real-life robots or highly realistic simulations or games. We believe that photorealism of observations are likely to help VLM annotations because a large portion of their training data comes from real-world photos or videos. For example, without providing an explicit context to a pretrained VLM, a camera image of a robot holding an object could probably be easier to interpret than a screenshot from gridworld environments or a Go board.
>
> While it’s true that in the abstract and simplified environments currently used for RL research, SENSEI might not always have an edge, we believe that as environments become richer and VLMs become more powerful there will be more and more potential applications for SENSEI.
>
> We thank the reviewer for raising this point and we now emphasize this more clearly in our discussion.
>
> ## Task specification in prompts
>
> > In my opinion, the actual prompt the authors use is simply a more specified version of that used in [1]
>
> We believe there might be some confusion regarding the types of prompts used in the Motif paper [ref 1]: Motif also uses environment specific knowledge to reduce noise in LLM annotations. Motif is only tested in Nethack and as default uses what the authors call “modifiers” in their prompts. In almost all of their experiments, the modifier contains environment specific knowledge, where the default modifier is:
>
> "Prefer agents that maximize the score in the game, for instance by killing monsters, collecting gold or going down the stairs in the dungeon."
>
> On top of that, Motif is uniquely situated as it is tested on NetHack: There is an abundance of publicly available NetHack wikis, guides etc. on the internet, which are likely also part of the LLM training data. Thus, Motif relies on the fact that the LLM is likely familiar with the game, which the authors also highlight in their paper.
>
> However, we do agree that the Robodesk prompt in our main experiments is likely over-specified and we could use less explicit, zero-knowledge prompts to train SENSEI. We illustrate this in a new experiment in Suppl D.4. Here, we annotated images using a multi-turn prompting strategy without prior information about the environment (details in Supp. C.3.2). First, we show a picture from the robotic environment of Robodesk, and ask the VLM for an environment description. Next, using this description in-context, we prompt the VLM to annotate pairs of images based on their interestingness. We analyze the reward function distilled from this general prompting strategy and show that it behaves qualitatively very similar to our original reward function, which used more environment-specific prompts, and matches the peaks in “interesting” moments of interactions.
>
> The reason we opted for a specialized prompt in our initial experiments is two-fold: First, similar to Motif, we wanted to increase annotation accuracy. Especially, since we are dealing with images that are not necessarily photorealistic and potentially out-of-distribution compared to the type of data VLMs are trained on. However, the second reason, which was our main concern, was due to practical limitations: Zero-knowledge prompts with multi-turn dialogues using GPT-4o cost double per annotation. With our specialized prompt, using a single-turn strategy, we were able to annotate 200K pairs for $400. For this new experiment, however, we could only annotate 100K pairs with this budget. As open-sourced VLMs get better, we expect multi-turn zero-knowledge strategies to prevail without cost constraints.

---

> > ### Author Response · Authors · 2024-11-24
> > **Response to Reviewer qRFT [2/2]**
> >
> > ## Double distillation
> >
> > > A more straightforward approach might be to incorporate the reward model directly into the world model
> >
> > Unfortunately, this suggestion is not possible. The RSSM, and other world models such as TD-MPC2 [ref 2], encode and predict dynamics fully in a self-learned latent state. This is crucial as the policy is trained in the world model’s imagination. Thus, for a world model to predict $r^\mathrm{sem}_t $ at any point in time $t$, we need a mapping from latent states to semantic rewards.
> >
> > Because the semantics of the latent state changes throughout training and across seeds, directly training Motif with latent state inputs is not feasible, as the same latent state can encode different contents at different points during training. Thus, we need to learn a mapping from latent states to semantic rewards.
> >
> > So the question is, is our mapping the best way to do it? Another option would be to decode the latent state to images and use those as inputs for Motif. However, we believe this has several disadvantages: 1) Decoding latent states to images is computationally costly which would significantly decrease our method’s computational efficiency. 2) This would still be a “double distillation” (latent state $\rightarrow$ images $\rightarrow$ $r^\mathrm{sem}$) only with an indirect target (image) instead of the direct target ($r^\mathrm{sem}$). 3) The image predictions of the RSSM can contain artifacts, blurriness or hallucinations. Since Motif is only trained on real images of the simulation, we will likely encounter out-of-distribution errors.
> >
> > We hope this motivates our “double distillation”. We thank the reviewer for allowing us to clarify this and we have now added a detailed explanation in the Supp. A.3.
> >
> > ## More Baselines
> >
> > Thank you for the suggestion to add more baselines. We have added Random Network Distillation (RND) [ref 3]  as another exploration baseline in Robodesk. RND is a popular exploration strategy that uses the prediction errors of random image embeddings as an intrinsic reward to train an exploration policy. In Robodesk, SENSEI explores substantially more interactions with all objects, except for button presses, than RND and as a result discovers on average much more task rewards.
> >
> > > There is only one baseline in the main experiment.
> >
> > We respectfully disagree with this simplification. We think both Plan2Explore and VLM-Motif both constitute interesting baselines for exploration. Additionally PPO and Dreamer are valuable baselines to benchmark the sample efficiency of learning a task-specific policy after exploration. Thus, SENSEI is compared against a number of strong baselines.
> >
> > We thank the reviewer again for their time and hope that we have addressed all concerns and misunderstandings and that we have answered all open questions.
> >
> > ## References
> >
> > [ref 1] Klissarov, Martin, et al. “Motif: Intrinsic Motivation from Artificial Intelligence Feedback”, ICLR 2024
> >
> > [ref 2] Hansen, Niklas, et al. “TD-MPC2: Scalable, Robust World Models for Continuous Control”, ICLR 2024
> >
> > [ref 3] Burda, Yuri, et al. "Exploration by random network distillation.", ICLR 2019.

---

> > > ### Comment · Reviewer_qRFT · 2024-11-26
> > >
> > > I appreciate the effort the authors have put into addressing the concerns raised during the review process.
> > >
> > > - While a camera image of real-life robots or highly realistic simulations may be easier for VLMs to interpret, this does not inherently guarantee a clearer expression of preferences between observations based on "interestingness" within this setup. I believe this response does not directly address my concern, and there remains a gap between the paper’s main claims and the experimental evidence provided.
> > > - I appreciate the authors conducting additional experiments with less specified prompts. This direction seems more appropriate, as defining "interestingness" specifically feels somewhat misaligned with the goal of extracting human priors about interestingness from VLMs. However, I find the statement that the results are “qualitatively very similar” or have a “high positive correlation” with the original reward function to be insufficient. Stronger conclusions would require demonstrating the advantages of using human priors in scenarios where they are indispensable or where performance significantly exceeds that achieved with the specified prompts.
> > > - The authors' explanation that Motif also relies on environment-specific knowledge does not meaningfully enhance the paper's contribution or novelty.
> > >
> > > While I am partially convinced by the authors’ clarifications regarding the formulation and baselines, my primary concern remains unaddressed. As such, my score remains unchanged.

---

> > > > ### Author Response · Authors · 2024-11-27
> > > > **New general prompt experiments and further clarifications on novelty and human priors**
> > > >
> > > > We appreciate that the reviewer likes our new less-specified prompting version. However, we believe there are still some points of confusion that we want to address.
> > > >
> > > > ## General prompting strategy
> > > > > „I appreciate the authors conducting additional experiments with less specified prompts. [...] However, I find the statement that the results are “qualitatively very similar” or have a “high positive correlation” with the original reward function to be insufficient.“
> > > >
> > > > We now train SENSEI with the reward function distilled from our general, less-specified prompting strategy. In Fig. 5 we show that this **more general version of SENSEI without external knowledge explores roughly as many object interactions as SENSEI with an external environment description, strongly outperforming Plan2Explore and RND in overall number of object interactions**. We believe this showcases the generality of our approach and that SENSEI does not rely on specific prompts.
> > > >
> > > > ## Human priors of interestingness
> > > > To avoid any misunderstanding about “human prior of interestingness”: We have taken this term from OMNI [1], referring to **human priors that are embedded into foundation models**.  Like OMNI, we consider foundation models that are trained on vast amounts of human-generated data to be a compression of human knowledge. We now made this clearer in our introduction.
> > > >
> > > > So, our "human notion of interestingness" is not referring to additional knowledge that is present in specific prompts. **And in our new experiment with general prompts, we showcase that environment-specific prompts are not necessary for SENSEI.**
> > > >
> > > > > „Stronger conclusions would require demonstrating the advantages of using human priors in scenarios where they are indispensable or where performance significantly exceeds that achieved with the specified prompts.“
> > > >
> > > > We suspect that your statement is based on a different interpretation of the notion of interestingness. See our answer above.
> > > > We do not claim that specific prompts are required nor are generally essential, but might be beneficial to steer the VLM. We argue that VLMs already contain pretty general knowledge useful in many tasks.
> > > > Specific prompts in our initial experiments were mostly required to keep the cost of annotations low.
> > > > > „The authors' explanation that Motif also relies on environment-specific knowledge does not meaningfully enhance the paper's contribution or novelty“
> > > >
> > > > We wanted to point out that your initial comparison to existing methods was inaccurate. Steering LLMs/VLMs via prompts is also tested in Motif and overall common practice in the literature [1, 2, 3], mainly due to existing limitations in foundation models. However, we agree with you that showcasing our method’s performance on more general prompts is important, which we have now added.
> > > >
> > > > ## Relation to Motif and Addressing Novelty
> > > > The main claim of our paper is the following: Distilling an interestingness signal of observations via prompting a VLM **combined with uncertainty-based intrinsic rewards** could lead to the exploration of more meaningful and useful high-level interactions, e.g., increased object interactions in Robodesk or key usages in MiniHack. We improve performance over popular exploration methods focused on uncertainty maximization or state coverage (e.g., Plan2Explore, RND).
> > > >
> > > > The information gain component is a key novelty of our method:
> > > > **Our combination of Motif with uncertainty-based exploration is completely novel.** Without epistemic uncertainty, Motif can get stuck at only certain interesting states with no incentive to explore further, as shown by our VLM-Motif ablation (Fig.4, Fig. 13). Original Motif tries to solve this problem by counting event occurrences during an episode, which is a strategy that 1) can only be applied for discrete settings, 2) does not scale to complex environments with continuous dynamics and high-dimensional observations like Robodesk.
> > > >
> > > > **We believe that especially the new version of SENSEI without environment-specific knowledge further increases the generality of our approach**,  together with the fact that compared to Motif, we only rely on visual observations and not text-based event captions provided by the environment. Additionally, unlike Motif, SENSEI does not rely on a large amount of human training data as the initial dataset for annotations. Instead SENSEI uses smaller datasets of observations collected through self-supervised uncertainty-based exploration.
> > > >
> > > > We hope these responses and our new results have clarified some misunderstandings about our claims and alleviated the reviewer’s concerns. We are happy to answer any further questions.
> > > >
> > > > ### References
> > > > [1] Zhang, J., et al. “OMNI: Open-endedness via Models of human Notions of Interestingness”, ICLR 2024
> > > >
> > > > [2] Klissarov, M., et al. “Motif: Intrinsic Motivation from Artificial Intelligence Feedback”, ICLR 2024
> > > >
> > > > [3] Du, Y., et al. "Guiding pretraining in reinforcement learning with large language models." ICML  2023.

---

> > > > > ### Author Response · Authors · 2024-12-02
> > > > > **End of discussion period**
> > > > >
> > > > > Dear Reviewer qRFT,
> > > > >
> > > > > we are checking in to see if you had an opportunity to read our new response and look at our new experimental results. If there are any remaining questions we are happy to answer those. If we have addressed your concerns satisfactorily, we kindly ask you to consider updating your score.

---

> ### Comment · Area_Chair_6YtS · 2024-11-25
> **Please read rebuttal**
>
> Dear Reviewer qRFT, Could you please read the authors' rebuttal and give them feedback at your earliest convenience? Thanks. AC

---

### Author Response · Authors · 2024-11-24
**General Response**

We thank all reviewers for their constructive feedback and appreciate that they found our research direction “well-motivated” and “valuable”, the method to be “straightforward and reasonable”, the experiments “diverse and convincing” with “ solid empirical validation”, and the paper “generally well-written”.

We notice that common suggestions among reviewers were more experimental evaluations, which we hope to address in this rebuttal. As a short summary, we provide 1) an ablation for self-supervised annotations using non-environment specific prompts, 2) a new exploration baseline (RND [ref 1]),  3) ablations on hyperparameter sensitivity and our novel coefficient adaptation strategy, 4) we provide more details on implementation, for example on the computational resources used, and 5) we ran more random seeds. We provide more details below and in the individual responses.

# Summary of changes

## General prompt with zero-knowledge for image annotation (qRFT & dZbp)

We investigated whether we can employ SENSEI using a more general prompting strategy without prior knowledge about the environment. In this general prompt setting (detailed in Suppl. C.3.2) we first prompt the VLM for an environment description given a screenshot and use this context to annotate image pairs with respect to their interestingness. In Suppl. D.4 we qualitatively analyze the resulting reward function for Robodesk and illustrate how the semantic rewards positively correlate or even match the ones obtained from our original, more specified, reward function.

## New baselines in Robodesk (qRFT & 71Ub & dZbp)

We’ve added Random Network Distillation (RND) [ref 1] with a PPO policy as a new model-free exploration baseline in Robodesk. SENSEI outperforms RND in almost all interaction metrics (Fig. 5) and discovered rewards (Fig 13), except for button presses. We also added VLM-Motif as a baseline in Robodesk, which SENSEI also outperforms in most interactions metrics (Fig. 15) and rewards (Fig. 13), showcasing the importance of the information gain objective of SENSEI.

## New ablations on hyperparameter sensitivity and dynamic coefficient adaption (71Ub)

We extensively analyze the effects of the dynamic coefficient adaptation strategy (Eq. 7) and the hyperparameter sensitivity of SENSEI in Robodesk in Suppl. D.6. In sum, the coefficient adaptation strategy makes our method much more robust to the exact choice of hyperparameters. Without coefficient adaptation, our method can still achieve a high performance for a subset of object interactions, but with coefficient adaptation, it achieves a high rate of interactions across the board.


## Computational resources (5Hj6 & dZbp)

We now detail our computational resources in a new supplementary section (Supp D.7).


## More seeds (71Ub)

We now run all Minihack experiments with 2 more seeds, so a total of 5 seeds. We will continue increasing the number of seeds for the camera-ready version, for both Minihack and Robodesk.

## PPO

By revisiting PPO, we also found an inconsistency in our implementation affecting PPO’s performance in MiniHack-KeyChest. We apologize for this. After re-running the experiments, PPO’s sample efficiency is increased in KeyChest. However, SENSEI still learns to reliably solve the task faster than PPO and, thus, the main message of these experiments remains unchanged.

We hope the detailed responses to the individual reviewers additionally address remaining questions. We thank all reviewers again for their time and efforts spent reviewing our paper.

## References

[ref 1] Burda, Yuri, et al. "Exploration by random network distillation.", ICLR 2019.

---

> ### Author Response · Authors · 2024-11-28
> **General Response 2: New experiments summarized**
>
> We thank all the reviewers for responding to our review and providing more feedback. To address concerns expressed in some of the responses, we provide two important updates to our paper.
>
> For the reviewer’s convenience, we highlight these newest changes of the paper in pink, while previous changes are highlighted in red.
>
> ## SENSEI with image annotations from general, zero-knowledge prompt
>
> In our first revision, we were able to distill a reward function using a more general prompting strategy without external knowledge about the environment by first prompting the VLM for an environment description and using this context for annotation (details in Suppl. C.3.2).
>
> We now run SENSEI with this general prompting setup in Robodesk. In Fig. 5 we show that this **more general version of SENSEI without external knowledge interacts roughly as often with relevant objects as our original SENSEI with an environment description generated by us**. Thus, even without external knowledge SENSEI clearly outperforms Plan2Explore and RND in overall number of object interactions. We believe this demonstrates how SENSEI’s performance does not hinge on specific prompts and these new results showcase the generality of our approach.
>
> ## Even more seeds
>
> We were able to increase the number of random seeds of our experiments. We now show MiniHack interactions with **10 random seeds** per configuration in Fig. 4. As expected, the previous trends hold and SENSEI outperforms the baselines in the number of rewards obtained, now also with reduced variance.
>
> We are currently rerunning all experiments of the main paper to increase the number of random seeds to 10 for all SENSEI configurations and baselines. Since we cannot modify the paper further, we will upload the updated plots (Fig. 5 & Fig. 6) to our supplementary website (linked in the paper, first footnote) once they are finished.
>
> We hope these changes address any remaining concerns about empirical evidence. We are happy to answer further questions.

---

### Meta-Review · Area_Chair_6YtS · 2024-12-19

**Metareview:**

This paper proposes a model-based RL framework that guides agents' exploration through human-defined "interestingness" from a vision-language model.  This work compares with existing baselines and shows superior performance. The core issue for this approach is the scalability of it. It can be challenging to find “interesting” parts purely from images.  Moreover, the overall method is complicated, which undermines the impact. Thus, I recommend the authors to polish the manuscript and submit to another conference.

**Additional Comments On Reviewer Discussion:**

Reviewer qRFT's concern is not addressed. Other concerns are addressed.

---

### Decision · Program_Chairs · 2025-01-22

Reject